# GrSMBMIP: Intercomparison of the modelled 1980-2012 surface mass balance over the Greenland Ice Sheet

Xavier Fettweis[1], Stefan Hofer[2], Uta Krebs-Kanzow[3], Charles Amory[1], Teruo Aoki[4,5], Constantijn J. Berends[6], Andreas Born[7,8], Jason E. Box[9], Alison Delhasse[1], Koji Fujita[10], Paul Gierz[3], Heiko Goelzer[6,11], Edward Hanna[12], Akihiro Hashimoto[4], Philippe Huybrechts[13], Marie-Luise Kapsch[14], Michalea D. King[15], Christoph Kittel[1], Charlotte Lang[1], Peter L. Langen[16], Jan T. M. Lenaerts[17], Glen E. Liston[18], Gerrit Lohmann[3], Sebastian H. Mernild[19,20,21,22], Uwe Mikolajewicz[14], Kameswarrao Modali[23], Ruth H. Mottram[16], Masashi Niwano[4], Brice Noël[6], Jonathan C. Ryan[24], Amy Smith[25], Jan Streffing[3], Marco Tedesco[26], Willem Jan van de Berg[6], Michiel van den Broeke[6], Roderik S. W. van de Wal[6, 27], Leo van Kampenhout[6], David Wilton[28], Bert Wouters[6,29], Florian Ziemen[14], Tobias Zolles[7,8]

[1]University of Liège, SPHERES research unit, Geography, Belgium.
[2]Department of Geosciences, Section of Meteorology and Oceanography, University of Oslo, Norway.
[3]Alfred-Wegener-Institut, Helmholtz Centre for Polar and Marine Research, Bremerhaven, Germany.
[4]Meteorological Research Institute, Japan Meteorological Agency, Tsukuba, Japan.
[5]National Institute of Polar Research, Tachikawa, Japan.
[6]Institute for Marine and Atmospheric research Utrecht, Utrecht University, The Netherlands.
[7]Department of Earth Science, University of Bergen, Bergen, Norway.
[8]Bjerknes Centre for Climate Research, Bergen, Norway.
[9]Geological Survey of Denmark and Greenland, Copenhagen, Denmark.
[10]Graduate School of Environmental Studies, Nagoya University, Nagoya, Japan.
[11]Laboratoire de Glaciologie, Université Libre de Bruxelles, Brussels, Belgium.
[12]School of Geography and Lincoln Centre for Water and Planetary Health, Lincoln, UK.
[13]Earth System Science & Departement Geografie, Vrije Universiteit Brussel, Brussels, Belgium.
[14]Max Planck Institute for Meteorology, Hamburg, Germany.
[15]Byrd Polar and Climate Research Center & School of Earth Sciences, The Ohio State University, Columbus OH, USA.
[16]Danish Meteorological Institute, Copenhagen, Denmark.
[17]Department of Atmospheric and Oceanic Sciences. University of Colorado Boulder, Boulder CO, USA.
[18]Cooperative Institute for Research in the Atmosphere, Colorado State University, Fort Collins, USA.
[19]Nansen Environmental and Remote Sensing Center, Bergen, Norway.
[20]Department of Environmental Sciences, Western Norway University of Applied Sciences, Sogndal, Norway.
[21]Geophysical Institute, University of Bergen, Bergen, Norway.
[22]Antarctic and Sub-Antarctic Program, Universidad de Magallanes, Punta Arenas, Chile.
[23]Universität Hamburg, Hamburg, Germany.
[24]Institute at Brown for Environment and Society, Brown University, USA.
[25]Department of Geography, University of Sheffield, UK.
[26]Lamont-Doherty Earth Observatory at Columbia University, New-York, USA.
[27]Department of Physical Geography, Utrecht University, Utrecht, the Netherlands.
[28]Department of Computer Science, University of Sheffield, UK.
[29]Department of Geoscience & Remote Sensing, Delft University of Technology, Delft, the Netherlands.

*Correspondence to*: Xavier Fettweis (xavier.fettweis@uliege.be)

**Abstract.** Observations and models agree that the Greenland Ice Sheet (GrIS) surface mass balance (SMB) has decreased since the end of the 1990's due to an increase of meltwater runoff and that this trend will accelerate in the future. However, large uncertainties remain, partly due to different approaches for modelling the GrIS SMB which have to weigh physical complexity or low computing time, different spatial and temporal resolutions, different forcing fields as well as

different ice sheet topographies and extents, which collectively make an inter-comparison difficult. Our GrIS SMB model intercomparison project (GrSMBMIP) aims to refine these uncertainties by intercomparing 13 models of four types which were forced with the same ERA-Interim reanalysis forcing fields, except for two global models. We interpolate all modelled SMB fields onto a common ice sheet mask at 1 km horizontal resolution for the period 1980-2012 and score the outputs against (1) SMB estimates from a combination of gravimetric remote sensing data from GRACE and measured ice

discharge, (2) ice cores, snow pits, in-situ SMB observations, and (3) remotely sensed bare ice extent from MODerate-resolution Imaging Spectroradiometer (MODIS). Spatially, the largest spread among models can be found around the margins of the ice sheet, highlighting model deficiencies in an accurate representation of the GrIS ablation zone extent and processes related to surface melt and runoff. Overall, polar regional climate models (RCMs) perform the best compared to observations, in particular for simulating precipitation patterns. However, other simpler and faster models have biases of

the same order as RCMs compared with observations and therefore remain useful tools for long-term simulations or coupling with ice sheet models. Finally, it is interesting to note that the ensemble mean of the 13 models produces the best estimate of the present day SMB relative to observations, suggesting that biases are not systematic among models and that this ensemble estimate can be used as a reference for current climate when carrying out future model developments. However, a higher density of in-situ SMB observations is required, especially in the south-east accumulation zone, where

the model spread can reach 2 mWE/yr due to large discrepancies in modelled snowfall accumulation.

## 1 Introduction

Mass loss from the Greenland Ice Sheet (GrIS) has been accelerating since the 1990s (Enderlin et al., 2014; Mouginot et al., 2019; Hanna et al., 2020; IMBIE2, 2020). Over the period 1992-2018, roughly 50% of the total mass loss can be ascribed to reduced GrIS surface mass balance (SMB) (IMBIE, 2020)


$$SMB = P - RU - SU - ER + GS \qquad (1)$$

which refers to the difference between the total precipitation (rain and snow, P), meltwater runoff (RU), sublimation/evaporation (SU), snow erosion by the wind (ER) and glacier storage (GS). Since drifting snow erosion

contributes ~1 Gt/yr (i.e. < 0.3%) to the SMB, ER is neglected in most models, although it can be important locally (Lenaerts et al., 2012). Moreover, the glacial water storage (supraglacial lakes, melt ponds, rivers) is neglected in this intercomparison, as it is not simulated by any model considered here. However, the mass changes coming from GS could be relevant when the SMB is integrated over the whole ice sheet, but have never been quantified until now.

Since the end of the 1990s, the models suggest that the surface melt has almost doubled, reaching record melt volume in the summers of 2012 and 2019, while the snowfall accumulation has remained approximately constant (Noël et al., 2019; Lenaerts et al., 2019; Tedesco and Fettweis, 2020). This recent GrIS SMB decrease - largely driven by the increase in meltwater runoff (Van den Broeke et al., 2016, Fettweis et al., 2017, Lenaerts et al., 2019, IPCC, 2019) - has been caused by Arctic amplification, a state change in the North Atlantic Oscillation, and increased Greenland Blocking events in summer (Fettweis et al., 2013b; Delhasse et al., 2018; Hanna et al., 2018, Hahn et al., 2020), which raise the average temperatures (Screen and Simmonds, 2010), reduce the cloudiness (Hofer et al., 2017) and enhance the melt-albedo feedback (Box et al. 2012; Ryan et al., 2019; Noël et al., 2019). While models agree well with satellite-based reconstructions, large uncertainties and model discrepancies remain in the current SMB evolution (IMBIE2, 2020). Additionally, SMB-related processes are one of the main uncertainties in future projections of the GrIS contribution to sea level rise as the ice sheet retreats in a warmer climate (Goelzer et al., 2013; van den Broeke et al., 2017; Hofer et al., 2019).

Therefore, there is a pressing need to improve and refine model estimates of recent SMB changes, for which we have in situ measurements and satellite data sets, in order to subsequently reduce the large model spread in future GrIS SMB projections. With the aim of reducing uncertainties in current modelled SMB estimates, we compare four types of SMB model, using 13 models in total to i) create an accurate multi-models SMB reconstruction over current climate and an associated uncertainty based on the ensemble of these models and ii) discuss the added values and drawbacks of each of them. Prior to this study, only a few attempts were made to compare the available models in terms of their ability to simulate the contemporary GrIS SMB (e.g. Vernon et al., 2013). These previous studies i) evaluated SMB within a subset of regional climate models (RCMs) (Rae et al., 2012), ii) compared positive degree day (PDD) models with energy balance snowpack models (van de Wal, 1996; Bougamont et al., 2007) or iii) assessed the representation of specific physical sub-processes (Reijmer et al., 2012). Since these models implement different physical and statistical processes, are run on different grids, use different forcing data, and/or cover various temporal ranges, previous model comparison studies suffer from limited inter-comparability.

In this study, we compare the SMB outputs of 13 state-of-the-art climate models (physical and statistical) over (1) a common time period (1980-2012), (2) using a common, 1 km spatial grid, and (3) over a common ice-sheet mask using the

contemporary GrIS extent. Moreover, 11 out of the 13 participating models are forced with ERA-Interim reanalysis (Dee et al., 2011), although each model prescribes the reanalysis forcing in a different manner.

Four kinds of models participate in our intercomparison:

- Positive-Degree Day (PDD) models using near-surface summer temperature to estimate melt and precipitation from forcing. The melt parameterisations of these models are relatively simple and due to the underlying assumptions they depend notably on the near-surface climate of their forcing (for precipitation in particular). However, the computational costs are very low and therefore they can be run at very high spatial resolutions and over long time scales.

- Energy Balance Models (EBMs) compute the surface energy balance to estimate melt by deriving surface energy fluxes and precipitation from forcing. Although the representation of surface melt is physically more robust than PDDs and they are able to simulate feedbacks including the melt-albedo feedback, they are also very dependent on the near-surface climate from the forcing data. However, similar to PDDs, EBMs are also computationally efficient. Therefore, both PDDs and EBMs are particularly useful to downscale large scale fields in the aim of

forcing ice sheet models over long periods .

- Regional climate models (RCMs) coupled with energy balance-based snow models compute energy fluxes, precipitation and the near-surface climate at a high resolution over the ice sheet. RCMs are forced at their lateral boundaries by global fields, mostly temperature, humidity and general circulation. A priori, they provide the best approach to represent the melt and precipitation patterns, as well as to simulate the surface-atmosphere

interactions at a high resolution. It is for this reason that the present SMB estimations of the Greenland ice sheet are mainly based on RCMs forced by reanalyses (IMBIE2, 2020). However, RCMs are computationally very expensive, limiting their simulations to typically 100 years. Finally, their results remain dependent on the biases in the forcing free-atmosphere fields above and around Greenland (Fettweis et al., 2013).

- General circulation models (GCMs) are global models that, unlike RCMs, have no spatial boundary conditions.

Instead, they require a small set of time-dependent primary inputs, such as aerosol emissions, greenhouse concentrations, and land use. Coupled with energy balance-based snow models, GCMs are capable of simulating GrIS SMB , which is particularly useful to perform future projections. However, to maintain reasonable computational time, their spatial resolution remains low, limiting their ability to explicitly simulate the snow-atmosphere interactions in the narrow ablation zone or the topography-induced precipitation. Coupled with an ice

sheet model, GCMs are the only tools that explicitly represent changes in general circulation in ocean or atmosphere, resulting from thinning of the ice sheet and other feedback processes.

For both PDDs and EBMs, the models are forced by the ERA-Interim near-surface climate extrapolated to the model's spatial resolution. In RCMs, the reanalysis dataset is prescribed at the ocean surface and at the lateral boundaries of their integration domain. Two types of GCM configurations are used in this study, (i) using an active ocean component, i.e. a truly free-running set-up, and (ii) an atmosphere-only configuration, where reconstructed historical sea-surface temperatures and sea ice cover are used as the boundary conditions over the ocean, possibly resulting in a modelled climate more closely resembling the real world (AMIP experiment, see Gates et al., 1998).

Sections 2 and 3 describe the 13 models used in the intercomparison (2 PDDs, 4 EBMs, 5 RCMs, and 2 GCMs) and the observational data sets used for the evaluation of these models. The models are inter-compared in Section 4 by highlighting the main discrepancies between models and evaluated in Section 5 with in-situ observations and satellite data. In Section 6, the discrepancies between models identified in Section 4 are linked to biases highlighted in Section 5 to propose the best estimates of the mean SMB and SMB changes over the current climate (1980-2012). Finally, conclusions are drawn in Section 7. Note that this intercomparison exercise aims to reduce uncertainty and identify regions with low measurement density and large model discrepancies in order to provide some insight on regional uncertainty; it is not the purpose here to formally rank model performance.

## 2. Model data

A brief description of each of the 13 models used in this study is provided in the following section and summarized in Table 1.

### 2.1 Description of the participating models

#### 2.1.1 BESSI (EBM – 10km)

BESSI is a surface energy and mass balance model designed for simulating long time scales (Born et al., 2019; Zolles et al., 2019). It is forced with ERA-interim reanalysis fields of temperature, humidity, long-wave and short-wave radiation, and precipitation (Dee et al., 2011). The temperature is the only variable that is downscaled to the actual model topography (ETOPO1, Amante and Eakins, 2009) using a lapse rate of 0.0065 K/m. Contrary to previously published model versions, here we use incoming longwave radiation as a forcing field rather than a temperature based parametrisation. Energy fluxes are calculated with a time step of one day on a 10x10 km grid.

The model uses an albedo scheme based on a linear relationship between temperature and a time decay rate (Aoki et al., 2003). This decay is enhanced in the presence of liquid water in the surface layer. The latent and sensible turbulent heat

fluxes are calculated based on the residual method (Rolstad and Oerlemans, 2005; Braithwaite, 2009) with constant wind speed over the entire ice sheet. Refreezing and percolation is instantaneous in every time step, with a maximum water holding capacity of 10% of the free pore volume (Greuell, 1992). Finally, the model parameters were optimised to fit the GRACE mass balance data over the 2002-2018 period (Born et al., 2019).

### 2.1.2 BOX13 (calibrated RCM – 5km)

The basis of the BOX13 surface mass balance reconstruction are linear regression parameters that describe relationships between spatially discontinuous in-situ records from meteorological stations (i.e. monthly temperature after Vinther et al. (2006); Cappelen et al. (2001, 2006, 2011) or firn/ice cores and spatially continuous outputs from the version 2.1 of the regional climate model RACMO (Ettema et al., 2010), described in Section 2.112. Explanatory (independent variable) data (air temperature and firn/ice core data) span 1840 to 2012. A 43-year overlap period 1960-2012 with RACMO2.1 is used to determine regression parameters on a grid cell basis. A fundamental assumption is that the calibration factors; regression slope and offset for the calibration period 1960-2012; is stationary in time.

The RACMO2.1 data are resampled and reprojected from a 0.1 deg (~10 km) grid to a 5 km grid. See Box et al. (2013) 'part I' for a description of the method, that includes a formal approach to estimate uncertainty. The following refinements are however made from the SMB reconstruction of Box et al. (2013) and Box (2013). The estimation of values is made for a domain that includes not only ice but land and sea. The physically-based meltwater retention scheme of Pfeffer et al. (1991) replaced the simpler approach used by Box (2013). Multiple station records contribute to the near surface air temperature for each given year, month and grid cell in the domain while in Box (2013), only data from the single highest correlating station yielded the reconstructed value. This revised surface mass balance data ends with year 2012 while Box (2013) ends in 2010. Finally, the annual accumulation rates from ice cores are dispersed into a monthly temporal resolution by weighting the monthly (based on the 1960-2012 RACMO2.1 data) fraction of the annual total for each grid cell in the domain. The accumulation reconstruction has been evaluated by Lewis et al. (2017, 2019).

### 2.1.3 CESM2 (GCM – 1km)

In this study, the CESM version 2.0 (CESM2) is used in a configuration with fixed ocean state. In particular, the protocol for the Atmospheric Model Intercomparison Project (AMIP, Gates et al., 1999) is used, with prescribed sea-surface temperatures and sea ice cover from Hurrell et al. (2008) for the period 1979-2014. Global land cover usage is also prescribed. The atmospheric and land components are the Community Atmosphere Model version 6 (CAM6) and the Community Land Model version 5 (CLM5), respectively, both operating at a nominal resolution of 1 degree. No ice dynamics are considered, i.e. the geometry of the GrIS is static in time. Initial conditions for CAM6 and CLM5 snow pack are

taken from a fully coupled CESM2 simulation. Subgrid topographic variability is partially accounted for by the use of multiple elevation classes (ECs) in CLM5, with up to 10 ECs per grid cell. Atmospheric forcing is downscaled to each EC, with lapse rates used for temperature and downwelling longwave radiation, and phase recomputation for precipitation (for details: see Van Kampenhout et al., 2020). Output indexed by EC is used for downscaling CESM2 SMB to the 1 km ISMIP6 grid (Nowicki et al., 2016) using linear interpolation in the vertical and bilinear interpolation in the horizontal direction. A detailed evaluation of present-day GrIS climate and SMB in CESM2 has been published in Van Kampenhout et al. (2020).

### 2.1.4 dEBM (EBM – 1km)

The diurnal Energy Balance Model (dEBM) is a surface mass balance scheme that incorporates both radiative and turbulent heat fluxes, and captures diurnal variability in the melt-freeze cycles (Krebs-Kanzow et al., 2018) and monthly variations in cloud cover. As forcing, dEBM only requires monthly means of short wave radiation at the surface, near surface air temperature and precipitation. Monthly mean duration and intensity of the diurnal melting and refreezing periods are derived from the monthly mean surface radiation and from the diurnal cycle of the top of atmosphere (TOA) short wave radiation. The latter is implicitly represented as a function of latitude and month based on prescribed parameters of the Earth's orbit around the Sun. Monthly mean atmospheric transmissivity and cloud cover are estimated from the ratio between monthly mean shortwave radiation at the surface and at the TOA (from forcing fields and from orbital parameters, respectively). The scheme has a monthly time step and distinguishes albedo of bare ice, and wet, dry and new snow on the basis of precipitation, surface energy balance and the previous month's snow type and snow height. Additionally, the scheme includes a residual heat flux R which is thought to represent those energy fluxes which are not included in the scheme, such as the heat flux to the subsurface or latent heat fluxes at the surface. Here, R has been treated as a tuning parameter and has been optimized to R = -5W m-2 with respect to the surface mass balance measurements from Machguth et al. (2016) over the ERA-Interim period (1979-2016). To force the model, monthly mean ERA-Interim precipitation, surface insolation and near surface air temperatures have been interpolated to the 1 km ISMIP6 grid and temperature fields have been additionally downscaled applying a lapse rate correction of $\Gamma$ = -0.007K/m.

### 2.1.4 HIRHAM (RCM – 5.5km)

The HIRHAM regional climate model has been developed to include a full surface energy and mass balance model using an original code developed from physical schemes used in the ECHAM5 global model and dynamical schemes from the HIRLAM numerical weather prediction model. It has 31 vertical levels and is forced on 6 hourly intervals on the lateral boundaries. The RCM has a simple five layer snowpack model to a depth of 10m over glacier surfaces, incorporating the same parameterisations used in an offline version that has 32 layers. The offline version assimilates MODIS MOD10A albedo data to get a closer fit between modelled and observed albedos. Langen et al. (2017) describe the snowpack model

in detail and show that the inclusion of MODIS data significantly improves the modelled SMB.

### 2.1.6 IMAU-ITM (EBM – 5km)

IMAU-ITM is an insolation- and temperature-based SMB model. This simplified EBM is used in the ANICE ice-sheet model for paleoclimate simulations (de Boer et al., 2014; Berends et al., 2018). Monthly precipitation from the ERA-Interim reanalysis is downscaled to actual model topography (in this case, the BedMachine v3 dataset; Morlighem et al., 2017) using the wind-orography-based parameterisation by Roe and Lindzen (2001) and Roe (2002). The resulting downscaled precipitation is partitioned into rain and snow based on the temperature parameterisation by Ohmura (1999). The depth of the accumulated snow layer is tracked, with a maximum value of 10 m; any additional firn is assumed to be compressed into ice. The surface albedo is calculated as a weighted average of the albedos of fresh snow and bare ice, based on the thickness of the snow layer and the amount of melt that occurred during the previous year. Melt is determined using the insolation-temperature parameterisation by Bintanja et al. (2002), which uses prescribed values for insolation at the top of the atmosphere, and which was developed especially for palaeoglaciological applications. Refreezing is calculated following the approach by Huybrechts and de Wolde (1999) and Janssens and Huybrechts (2000), based on the available liquid water (the sum of rain and melt) and the refreezing potential, integrated over the entire year to account for the retention of summer melt which is refrozen in winter. For this study, the parameters in the refreezing and snowmelt parameterisations were calibrated to obtain the closest match (i.e. highest value of linear correlation coefficient divided by RMSE) to the RACMO2.3 values over the 1979-2017 period on the 1 km grid.

### 2.1.7 MAR (RCM - 15km)

The version 3.9.6 from MAR is used here by using a resolution of 15 km. MAR was forced at its lateral boundaries by ERA-Interim at a 6-hourly time step. The boundary forcing files include information about the temperature, u- and v- wind components, specific humidity and sea level pressure as well as the sea surface temperature (SST) and sea ice cover over ocean. It is the same model configuration which is used in the Ice Sheet Model Intercomparison Project for CMIP6 (ISMIP6) for future projections over the GrIS (Nowicki et al., 2016). With respect to the version 3.5.2 of MAR used in Fettweis et al. (2017) and Hofer et al. (2017), the main improvements are: (1) An increase of the cloud lifetime with the aim of correcting the biases of solar and infrared radiation highlighted in Fettweis et al. (2017), (2) adjustments in the bare ice albedo representation for a better comparison with in situ measurements, (3) a larger independence of model results to the used time step and (4) a better dynamical stability with an increased spatial filtering for a computing time divided by a factor 2 compared to version 3.5.2. Additionally, we also dealt with minor bug corrections and small updates for enhanced computing efficiency and comparison with in-situ automatic weather data (Delhasse et al., 2019).

### 2.1.8 MPI-ESM (GCM - 1km)

The historical simulation underlying the SMB calculations by Max Planck Institute (MPI) is simulated with a higher resolution version of the latest version of the MPI Earth System Model (MPI-ESM1.2-HR). In this version the atmospheric model ECHAM6.3, with a spectral resolution of T127 (~100 km), is coupled to the ocean model MPIOM version 1.6.2, with a nominal 0.4° resolution and a tripolar grid. A thorough description of this model setup can be found in Müller et al. (2018). An EBM approach is used to calculate the SMB from one ensemble member of the historical MPI-ESM1.2-HR simulations (Mauritsen et al., 2019) and downscale it from ~100km to the 1km ISMIP6 topography. The offline EBM scheme is similar to the one presented in Vizcaíno et al. (2010); despite technical changes and the introduction of elevation classes mainly the albedo parameterisation was updated. The EBM calculates melt and accumulation rates from hourly atmospheric fields of the historical MPI-ESM1.2-HR simulation on its native grid. The atmospheric fields are bi-linearly interpolated onto 24 fixed elevation classes, ranging from 0 m to 8000 m. To account for height differences between each elevation class and the surface elevation of the atmospheric model a height correction is applied to near-surface air temperature, humidity, dew point temperature, precipitation, downward longwave radiation and near-surface density fields. The downward shortwave radiation is kept constant, as it is largely affected by atmospheric properties that are independent of elevation differences (e.g. ozone concentration, aerosol thickness) (Yang et al., 2006). To obtain melt rates, the EBM computes the energy balance at the atmosphere-snow interface as the sum over the radiative and turbulent as well as rain induced and conductive heat fluxes. The albedo parameterisation used here is based on the parameterisation by Oerlemans and Knap (1998) and considers snow ageing, snow depth, and the influence of cloud coverage. The obtained 3-D fields of surface melt, accumulation and SMB are then vertically and horizontally interpolated onto the 1km ISMIP6 topography used as reference topography in this study.

### 2.1.9 NHM-SMAP (RCM – 5km)

The latest version of the polar RCM NHM-SMAP, with a horizontal resolution of 5 km, developed by Niwano et al. (2018) was used in this study. The same version was recently utilised to assess cloud radiative effects on the Greenland ice sheet surface melt (Niwano et al., 2019). The atmospheric part of NHM-SMAP is the Japan Meteorological Agency Non-Hydrostatic atmospheric Model (JMA-NHM) developed by Saito et al. (2006), which employs flux form equations in spherical curvilinear orthogonal coordinates as the governing basic equations. We pay close attention to the cloud microphysics processes, therefore, the version of JMA-NHM utilised for NHM-SMAP (Hashimoto et al., 2017) employs a double-moment bulk cloud microphysics scheme to predict both the mixing ratio and the concentration of solid hydrometeors (cloud ice, snow, and graupel), and a single-moment scheme to predict the mixing ratio of liquid hydrometeors (cloud water and rain). For the simulation of snow and ice physical conditions, the multilayered physical snowpack model SMAP is utilised (Niwano et al., 2012, 2014). The SMAP model calculates snow albedo using the detailed

physically based snow albedo model developed by Aoki et al. (2011) considering the effects of snow grain size evolution explicitly. Although the model can also consider the effects of light-absorbing impurities on snow albedo, we assumed the pure snow condition here. On the other hand, bare ice albedo is calculated by using a simple parameterisation as a function of density. To estimate realistic runoff from the ice sheet, a detailed vertical water movement scheme based on the Richards equation (Yamaguchi et al., 2012) is used. To force NHM-SMAP (dynamical downscaling), we used not the ERA-Interim reanalysis but the JRA-55 reanalysis (Kobayashi et al., 2015) due to the lack of enough computational resources. However, it should be noted that the quality of the arctic atmospheric physical conditions from both reanalysis data during the study period were almost the same level as reported by Simmons and Poli (2015) and Fettweis et al. (2017) who showed no significant difference between MAR forced by ERA-Interim and JRA-55.

### 2.1.10 PDD5km (PDD - 5km)

European Centre for Medium-Range Weather Forecasts (ECMWF) ERA-Interim (Dee et al., 2011) 2-m surface air temperature, precipitation and surface latent heat flux reanalysis data were downscaled from their native 0.75° resolution to 5x5-km using bilinear interpolation, a high-resolution DEM (Ekholm, 1996) and empirically-derived ice-sheet surface lapse rates to correct surface air temperature, as described in full in Hanna et al. (2005, 2011). Downscaled surface air temperature was validated using independent in-situ observational automatic weather station data from the Greenland Climate Network (Steffen and Box, 2001), showing very good agreement between downscaled/modelled and observed temperatures. Net solid precipitation (snowfall minus evaporation and sublimation) was spatially calibrated against the Bales et al. (2009) kriged map of snow accumulation based on ice-core and coastal precipitation gauges. Evaporation and sublimation were calculated from surface latent heat flux. The resulting downscaled Greenland climate gridded data were used to drive a runoff/retention model (Janssens and Huybrechts, 2000) that produced surface melt, runoff, evaporation and SMB at a monthly time resolution, while net precipitation was taken from the ERA-I dataset downscaled, calibrated and adjusted as above. Ice-sheet averaged annual SMB since 1958 was shown to correlate strongly between this method and RACMO2.1 but significant differences in absolute values between the respective methods were considered to be mainly due to poorly-constrained modelled accumulation (Hanna et al., 2011).

### 2.1.11 PDD1km (PDD - 1km)

The modelling method is essentially the same as described in 2.1.10. However, here a higher-resolution DEM (Bamber et al., 2013) was used to downscale ERA-Interim reanalysis data to 1x1 km2 resolution, producing monthly output for 1979-2012 (Wilton et al. ,2017). In addition, variable "sigma" - standard deviation of 6-hourly temperatures, computed for each month- was incorporated into the PDD method, based on earlier work by Jowett et al. (2015). The resulting high-resolution PDD model output was evaluated using PROMICE observations (Machguth et al., 2016), showing generally robust

correlations (Wilton et al., 2017) which were broadly comparable, though not quite as good, as the polar RCMs. Finally, this method is particularly useful for long centennial/pre-satellite timescales for which relatively few reliable meteorological fields are available (Wilton et al., 2017).

### 2.1.12 RACMO2.3 (RCM - 1km)

The polar (p) version of the Regional Atmospheric Climate Model (RACMO2.3p2) is run at 5.5 km horizontal resolution for the period 1958-2018 (Noël et al., 2019). The model incorporates the dynamical core of the High-Resolution Limited Area Model (HIRLAM; Undèn et al., 2002) and the physics from the European Centre for Medium-range Weather Forecasts-Integrated Forecast System (ECMWF-IFS cycle CY33r1; ECMWF-IFS, 2008). RACMO2.3p2 includes a multi-layer snow module that simulates melt, water percolation and retention in snow, refreezing and runoff (Ettema et al., 2010). The model also accounts for dry snow densification (Ligtenberg et al., 2018), and drifting snow erosion and sublimation (Lenaerts et al., 2012). Snow albedo is calculated based on snow grain size, cloud optical thickness, solar zenith angle and impurity concentration in snow (Van Angelen et al., 2012). Bare ice albedo is prescribed from the 500 m MODIS 16-day Albedo product (MCD43A3), as the 5% lowest surface albedo records for the period 2000-2015, minimized at 0.30 for dark bare ice and maximized at 0.55 for bright ice under perennial firn. Glacier outlines and surface topography are prescribed from a down-sampled version of the 90 m Greenland Ice Mapping Project (GIMP) Digital Elevation Model (DEM) (Howat et al., 2014). RACMO2.3p2 is forced at its lateral boundaries by ERA-40 (1958-1978) (Uppala et al., 2005) and ERA-Interim (1979-2018) (Dee et al., 2011) re-analyses on a 6-hourly basis within a 24 grid cells wide relaxation zone. The forcing consists of temperature, specific humidity, pressure, wind speed and direction being prescribed at each of the 40 vertical atmosphere model levels. Upper atmosphere relaxation (nudging) is also implemented in RACMO2.3p2 (Van de Berg and Medley, 2016). The model has 40 active snow layers that are initialized in September 1957 using temperature and density profiles derived from the offline IMAU Firn Densification Model (IMAU-FDM) (Ligtenberg et al., 2018). Detailed description of the model and recent updates are discussed in Noël et al. (2018, 2019).

The 5.5 km product is further statistically downscaled onto a 1 km grid to resolve the steep SMB gradients over narrow glaciers and confined ablation zones at the rugged ice sheet margins. Statistical downscaling corrects runoff for biases in elevation and bare ice albedo using a down-sampled version of the GIMP DEM (topography and ice mask) and a MODIS albedo product at 1 km resolution. This allows to accurately represent the high runoff rates observed at the GrIS margins, significantly improving the agreement with SMB measurements. Detailed description of the statistical downscaling procedure is discussed in Noël et al. (2016).

### 2.1.13 SnowModel (EBM - 5km)

SnowModel was forced with ERA-Interim (ERA-I) reanalysis products on a 0.75° longitude × 0.75° latitude grid from the

European Centre for Medium-Range Weather Forecasts (ECMWF; Dee et al. 2011), where the 6-hour (precipitation at 12-hour) temporal resolution ERA-I data were downscaled to 3-hourly values and a 5-km grid. SnowModel (Liston and Elder, 2006a) contains six sub-models, where five of the models were used here to quantify spatiotemporal variations in

atmospheric forcing, GrIS surface snow properties (including refreezing and retention), sublimation, evaporation, runoff, and SMB. The sub-model MicroMet (Liston and Elder, 2006b; Mernild et al., 2006) downscaled and distributed the spatiotemporal atmospheric fields using the Barnes objective interpolation scheme, where the interpolated fields were also adjusted using known meteorological algorithms, e.g., temperature-elevation, wind-topography, humidity-cloudiness, and radiation-cloud-topography relationships (Liston and Elder, 2006b). Enbal (Liston, 1995; Liston et al., 1999) simulated a

full surface energy balance considering the influence of cloud cover, sun angle, topographic slope, and aspect on incoming solar radiation, and moisture exchanges, e.g., multilayer heat- and mass-transfer processes within the snow (Liston and Mernild, 2012). SnowTran-3D (Liston and Sturm, 1998, 2002; Liston et al., 2007) accounted for the snow (re)distribution by wind. SnowPack-ML (Liston and Mernild, 2012) simulated multilayer snow depths, temperatures, and water-equivalent evolutions. HydroFlow (Liston and Mernild, 2012) simulated watershed divides, routing network, flow residence-time, and

runoff routing (configurations based on the hypothetical gridded topography and ocean-mask datasets), and discharge hydrographs for each grid cell including from catchment outlets. These sub-models have been tested against independent observations with success in Greenland, Arctic, high mountain regions, and on the Antarctic Ice Sheet with acceptable results (e.g., Liston and Hiemstra, 2011; Mernild and Liston, 2012; Mernild et al., 2015; Beamer et al., 2016).

**2.2 Interpolation on a common grid**

One of the key issues raised by the first SMB model intercomparison performed by Vernon et al. (2013) was the high dependency of modelled integrated SMB values to the used ice sheet mask. To mitigate this problem, we interpolate all model outputs to the same 1-km grid used in the Ice Sheet Model Intercomparison Project for CMIP6 (ISMIP6). This resolution is chosen because the highest resolution model outputs (e.g. RACMO2.3p2) are available at 1 km and choosing a coarser resolution could compromise their quality. A common grid also allows a comparison on two common ice sheet

masks: the contiguous Greenland ice sheet, which is common to all the models and the Greenland ice sheet plus peripheral ice caps and mountain glaciers, common to all the models except the two PDD models. Unless otherwise indicated, the SMB components have been interpolated to 1 km using a simple linear interpolation metric of the four nearest inverse-distance-weighted model grid cells. Moreover, as done in Le Clech't et al. (2019), the interpolated 1-km SMB and runoff fields have been corrected for elevation differences between the model native topography and the GIMP 250 m

topography (upscaled to 1 km here), using time and space-varying SMB-elevation gradients, similar to Franco et al. (2012) and Noël et al. (2016). No correction was applied to precipitation after interpolation to 1-km. Finally, the ensemble mean is based on the average of the 13 modelled monthly outputs interpolated onto the common 1-km grid.

### 3. Observational data

#### 3.1. Ice core and SMB measurements.

Similar to Fettweis et al. (2017), we compare modelled SMB with in-situ observations from:

(1) ice core measurements in the accumulation zone (Bales et al., 2001, 2009; Ohmura et al., 1999). The model outputs are averaged over the overlapping measurements period. We use the annual mean over 1980-2012 when the measurements
period is not specified or not included into the period 1980-2012, to increase the number of available ice core measurements for model evaluation, as the snowfall accumulation has remained approximately constant over the last decades (Fettweis et al., 2017). The modelled SMB values are compared to ice cores by interpolating the four nearest inverse-distance-weighted grid cells to the common 1-km ISMIP6 grid.


(2) airborne radar transects in the accumulation zone from Karlsson et al. (2016, 2020). For all of these measurements, the annual mean over 1980-2012 is used as comparison and as for the ice cores, outputs are interpolated using the four nearest inverse-distance-weighted grid cells to the common 1-km ISMIP6 grid.

(3) the SMB database (Machguth et al., 2016) compiled under the auspice of PROMICE and available through the PROMICE web portal (http://www.promice.dk). This dataset mainly covers the ablation zone of the GrIS and includes measurements over some peripheral ice caps (as shown in Fig 1). Measurements not included in the 1980-2012 period, records shorter than 3 months or located outside the common 1 km ice mask are discarded from the comparison. In a similar fashion as in Wilton et al. (2017), monthly model outputs are weighted by the length of the observed month, e.g. if the record starts in
the middle of a month. Daily outputs, available for some models, are not used here and as for the ice cores, outputs are interpolated using the four nearest inverse-distance-weighted grid cells onto the ISMIP6 ice mask.

(4) the unpublished database of snow pits (Jason Box, personal communication) incorporating observations of winter accumulation over previously exposed bare ice or firn. Snow pits were monitored at the end of the following winter
accumulation period (usually in May). As only the date when the snow pits were dug is known (May), we assume, for the comparison with the models, that each record has started on the 1st of September. However, for some years and locations, the winter accumulation may have started slightly later in October or November, after some late-season melt events. That is why, we have accumulated modelled SMB values from September to May when the monthly modelled SMB is positive.

### 3.2. GRACE estimation

The GRACE-based product, coupled with an estimate of monthly ice discharge from all (n > 200) large outlet glaciers (King et al., 2018), is used here to evaluate the trend of the 2003-2012 modelled SMB. These quantities are integrated over the 6 basins defined in King et al. (2018) and based on basin configurations from Sasgen et al. (2012). The correction for glacial isostatic adjustment is based on the model of Khan et al. (2016). Finally, in King et al. (2018), monthly glacier discharge estimates were combined with RACMO2.3p2 based SMB, and compared to the resulting total mass balance estimate from the GRACE product as will be done in this study for each model.

### 3.3. Bare ice extent

The MODIS-based bare ice monthly product was used to evaluate the mean extent of the ablation zone (i.e. where the mean annual SMB is negative) simulated by the models over 2000-2012 (Ryan et al., 2019). The daily classified bare ice maps were used to calculate a summer (June, July, and August) bare ice presence index (or exposure frequency). The bare ice presence index varies between 0 and 1 in any given summer and is defined as the number of times a pixel is classified as bare ice divided by the total number of valid observations of that pixel (i.e., when not cloud obscured) between June 1 and August 31. Finally, a 1x1 km² pixel was considered within the ablation zone if it was detected as bare ice in at least 50% of the summers in 2000-2012.

### 4. Model intercomparison

Integrated over the common main ice sheet mask (see Table 2), the average total GrIS SMB over 1980-2012 ranges from 96 Gt/yr (SnowModel) to 429 Gt/yr (NHM-SMAP), with a mean value of ~ 340 +/- 110 Gt/yr. Comparing the two largest SMB components (i.e. snowfall and runoff), we can see large discrepancies between models. For some models, SMB falls within the range of the other models only due to compensating effects of over- or underestimating both snowfall and runoff (see supplementary Fig. S1). For example, the snowfall and runoff from BESSI and the PDD models are very low compared to other models but yield similar integrated SMB values. In addition, the SMB of NHM-SMAP (resp. SnowModel) is substantially higher (resp. lower) than that of other models, due to larger snowfall accumulation (resp. meltwater runoff) than other models. Except for SnowModel (which suggests a SMB trend close to -12.9 Gt/yr²) and both GCMs, all models suggest that the SMB of the GrIS has decreased at a rate of ~7 Gt/yr² over the period 1980-2012, primarily due to an increase of meltwater runoff (~ +8 Gt/yr²). It is also interesting to note that the meltwater runoff increase is about 2 times lower in the GCMs, probably because a recent increase of atmospheric blocking events in summer, increasing surface runoff, is not captured by the GCMs (Hanna et al., 2018). Finally, while Bougamont et al. (2007) concluded that PDDs are more sensitive to climate warming than the EBMs, it is not the case here as the PDD based melt rates are fully included in

the EBMs-based spread, including over the extreme summers (e.g. 2012).

If we compare each model to the ensemble mean (see Figs. 1 and 2), we can see that BESSI, NHM-SMAP and PDD1km generally simulate lower runoff in the ablation zone compared to the other models (Fig. 3). In contrast, SnowModel, HIRHAM and BOX13 simulate larger runoff than the ensemble mean. These differences largely explain the SMB anomaly in the ablation zone shown in Fig. 2 with respect to the ensemble mean. Finally, IMAU-ITM, BESSI and SnowModel are drier in the interior of the ice sheet (Fig. 4), even though they use identical ERA-Interim precipitation forcing as the other PDD/EBM

models.

In the accumulation zone of South Greenland, CESM simulates larger snowfall rates while MPI-ESM simulates small ones in addition to larger runoff than the ensemble mean. Finally, for all the RCMs, the snowfall accumulation does not show similar and systematic deviations over a large extent from the ensemble mean. This better representation of the spatial

variability in precipitation in the RCMs is likely due to the fact that the precipitation is resolved at higher resolution than in both the GCMs and the ERA-Interim reanalysis, the latter of which is used to force the PDD/EBM models. This highlights the advantage of simulating precipitation at high spatial resolution in order to represent the interaction between the atmospheric flow and (ice-sheet) topography. The south-east coast of Greenland shows the largest discrepancy between models, reaching 2 mWE/yr locally, and this is where most RCMs simulate higher precipitation than other types of models.

Unfortunately, in-situ data coverage along the south-eastern coast is very sparse, making it hard to prove whether high accumulation rates in RCMs, locally exceeding 3 mWE/yr, are actually realistic. This highlights the need for a higher density of in-situ measurements in southeast Greenland where the models simulate the maximum of precipitation. Shallow ice radar or remote sensing ( elevation changes) could also help to evaluate the accumulation rates in this area.

## 5. Evaluation of models

### 5.1 Comparison with in-situ SMB measurements

In comparison with SMB derived from ice cores (location shown in Fig. 5), both PDD models perform the best (See Table 3). However, the same ice core dataset has been used to correct ERA-interim precipitation as that used to force both of these models, therefore they are not completely independent. Furthermore, the RCMs MAR, NHM-SMAP and RACMO2.3 generally agree better with observations than the other models that use ERA-Interim precipitation as forcing or GCM-based

precipitation computed at lower spatial resolution. Except for the two PDD models, the RMSE of the models is generally larger than the standard deviation of the ice core measurements meaning that the model biases are statistically significant.

In comparison with airborne radar transects, all the models compare very well and biases are not significant but these transects are representative of only a small part of the dry snow zone, already covered by some ice core measurements. Finally, as for the ice cores, the best comparison occurs for the PDD models. We can also note that the correlations are lower for both GCMs as a result of a coarser spatial resolution in GCMs disallowing the representation of the spatial variability of SMB. However, the RMSE from GCMs is comparable to the other models.

All the models show a worse agreement with the 130 snow pits than with the ice cores measurements (Table 3). However a large part of these discrepancies can likely be ascribed to the use of monthly outputs knowing that the starting date of the snow pit records (i.e. when the winter accumulation actually started) is uncertain. With respect to the PROMICE SMB data set, the model RMSE varies between 0.48 mWE (for MAR) and 0.89 mWE (for BESSI) over the main ice sheet. For most of the models, the RMSE is close to the temporal standard deviation (0.62 mWE) of the PROMICE data set, suggesting that the modelled biases are not statistically significant. Finally, it is interesting to note that the best statistics are performed with the ensemble mean of the 13 models (see Fig. S1 in supplementary), which will be used hereafter as the SMB reference field. This suggests also that biases of each model are of different signs and are compensated when the 13 models based estimates are averaged.

In Fig 5, we can see that all models underestimate most of the few measurements that we have above 1 mWE and underestimate ablation rates greater than 3 mWE, except RACMO2.3 and the two PDD models. Between 0 and 2 mWE, most of the models rather overestimate ablation. Finally, BESSI, BOX13 and NHM-SMAP systematically underestimate the ablation rates over the whole range of observations, explaining their unfavourable statistics in Tab. 3 relative to other models.

In brief, it is interesting to note that all types of model generally show similar performance (see Fig. S2 in supplementary material). Computationally expensive models (i.e. the RCMs) give the best agreement with observations: MAR and RACMO2.3 perform very well on average compared to SMB observations GrIS-wide, while NHM-SMAP (resp. HIRHAM) performs better at representing SMB in the accumulation zone (resp. in the ablation zone). However, the evaluation statistics from the more simple models (PDDs and EBMs) and from GCMs are generally similar to the ones of polar RCMs. It is nevertheless important to note that RCMs were used to calibrate some of these models, partly explaining their general good performance.

## 5.2 Comparison with GRACE measurements

To enable comparison with GRACE mass change, we estimate total mass balance (MB) for each model in 6 basins (see Fig. 6

and Table 4) by subtracting observed ice discharge D (King et al., 2018) from modelled SMB for the 13 models, as King et al. (2018) originally did using the RACMO-based SMB:


$$MB = SMB - D \qquad (2)$$

The GRACE signal variability is mainly a combination of the i) seasonal cycle (accumulation in winter and melt in summer), ii) interannual variability in this seasonal cycle and, iii) long- climate variability (linked in part to global warming) induced

mass loss, that we assume here to be the linear trend over the considered period (2003-2012). Over Basin 1 and 2, IMAU-ITM and SnowModel (resp. CESM2 and MPI-ESM) overestimate (resp. underestimate) the mass loss in the GRACE signal (i.e. the linear trend) over 2003-2012. Over Basin 3, all the models underestimate the mass change. Additionally, some of the models (in particular MAR) do not simulate mass variations in Basin 3, despite GRACE data suggesting a mass loss of 450 Gt over 2003-2012. In south Greenland (Basin 4), the two PDD models show the most favourable statistics but all the

models (except MPI-ESM) underestimate mass loss. Along the west coast (Basin 5 and Basin 6), MAR and RACMO2.3 are most closely aligned with the observations, while SnowModel systematically overestimates, and NHM-SMAP systematically underestimates the mass loss. For the other models, the bias in Basin 5 and 6 varies in sign. Finally, an EBM (dEBM), a GCM (MPI-ESM), a PDD (PDD1km) and two RCMs (MAR and RACMO2.3) compare the closest to the GRACE-derived GrIS-integrated mass loss over 2003-2012. These favourable statistics are due to error compensation as none of the models

matches well with the GRACE-derived regional mass loss integrated over individual basins.

For a total surface mass loss over 2003-2012 of ~ 3000 Gt as suggested by the GRACE data set, the models range from -1066 Gt to -6034 Gt with an ensemble mean of -2611 +/- 1253 Gt suggesting a large discrepancy between models and therefore a large uncertainty in the modelled SMB trends over the current climate. It is nevertheless interesting to note

that, for most of the models, the sign of the bias when compared to the PROMICE data (see Table 3) is highly correlated to the sign of the trend bias with respect to the GRACE based product. For example, as the 2003-2012 changes in SMB were driven by an increase of melt, those models underestimating surface ablation also underestimate these recent changes (the signal coming from discharge change is the same for all the models). However, we need to mention that all of our modelled total mass balance estimations use the same discharge estimates from King et al. (2018) and do not take into

account changes in mass over tundra, over small ice caps (not included in the common ice sheet mask) and in glacial storages (e.g. meltwater lakes, water tables,...): this partly explains why the discrepancies between models and GRACE could be very high in some areas.

Finally, by removing the linear trend in both time series (i.e. the signal mainly coming from the surface melt increase over

2003-2012 as no change in snowfall accumulation is suggested by the models) we are able to evaluate the seasonal/inter-annual variability gauged here by the RMSE and the correlation listed in Table 4. We find that the five RCMs simulate the seasonal cycle of SMB much better than other types of models (see supplementary Fig. S3), although the linear trend over 2003-2012 (Fig. 6) is significantly underestimated in e.g. HIRHAM and NHM-SMAP. The two GCMs have a larger RMSE mainly because they are not forced by ERA-Interim, hence, years with large/low snowfall accumulation/melt do not

necessarily take place at the same time in the GCMs as in the real climate; still the linear trend over 2003-2012 compares very well with GRACE (e.g. for MPI-ESM).

## 5.3 Comparison with bare ice extent

We can reasonably assume that the mean SMB should be negative in the bare ice area and positive above the snow line. However, the equilibrium line altitude varies each year. Therefore, we have chosen to only use SMB values that fall within

0 mmWE/yr plus (resp. minus) half the SMB interannual variability (/2) to detect the modelled accumulation (resp. ablation) zone. Except BESSI, all models are able to develop a large enough bare ice area, although most of them overestimate the ablation zone extent, in particular IMAU-ITM and SnowModel (see Table 5). In Fig. 1, the hatched areas outline the regions where the models overestimate or underestimate (only for BESSI) the bare ice area. We can see that BESSI fails to represent the extent of the south-western ablation zone. Conversely, IMAU-ITM, BOX13, PDD5km and

SnowModel overestimate the extent of the ablation area in north-east Greenland, where the SMB from the other models is also very low but remains positive. Finally, it is interesting to note that both GCMs (CESM2 and MPI-ESM), despite their coarse native resolution in the atmosphere, are able to accurately model the mean snow line, which is attributed to their subgrid downscaling module.

## 6. Discussion

After having inter-compared the models in Section 4 and evaluated them with in situ and satellite observations in Section 5, this section aims to link the results discussed in the two previous sections. In Fig. 7, we can see that the deviation with the GRACE trend is roughly a linear function of the annual mean GrIS SMB over 1980-2012 and of this trend. The models under/over-estimating meltwater runoff with respect to the ensemble mean (which compares well with GRACE), systematically under/over-estimate the GRACE-derived mass loss, largely driven by the increase of meltwater runoff in

agreement with the past assessments of SMB seasonal variability using GRACE (Velicogna et al., 2014; Alexander et al., 2016; Schlegel et al., 2016). As the sensitivity of the runoff to temperature increase is not linear (Fettweis et al., 2013), the runoff increase is lower/larger for the models under/over-estimating the meltwater runoff. With respect to GRACE, the best comparisons occur for mean SMB rates between ~ 280 and ~ 380 GT/yr over 1980-2012 and SMB trends between ~ - 9

and ~ -6 GT/yr² over 1980-2012. With respect to the PROMICE data set, the best comparisons occur for mean SMB rates (resp. SMB trend) between ~ 280 and ~ 380 GT/yr over 1980-2012 in agreement with GRACE (resp. between ~ - 9 and ~ -7 GT/yr²). According to the linear regression line in Fig.7, the best mean SMB and SMB trend estimates are ~ 320 GT/yr and ~ − 7.2 GT/yr², which are very close to the values of the ensemble mean of the 13 models. This suggests that the ensemble mean can be reliably chosen as the best reference to represent the mean SMB and its variability over 1980-2012 for any validation of future model developments.

Except the two GCMs that underestimate the recent surface mass loss trend mainly because they have not the same general circulation variability than the ERA-interim climate, the runoff anomaly expressed in percentage of the ensemble mean remains mainly constant in time, including the extreme summer 2012. This first confirms that the modelled runoff overestimation/underestimation is systematic over time, independently of the physics used in the model. This also suggests that a runoff bias over current climate should increase in absolute value for warmer climates in the same proportion than runoff, justifying the importance of well representing the current mean climate and trend before performing future projections.

Finally, for 95% (resp. 90% and 99%) of the 10767 in-situ measurements (see Fig. 1) over the main ice sheet, the mean bias between the ensemble mean and the measurements represents 72% (resp. 70% and 75%) of the model spread around this ensemble mean; it also corresponds to 16% (resp. 15% and 18%) of the observed values. Therefore, we can reasonably estimate three quarters of the model spread around the ensemble mean as the uncertainty of this ensemble mean based SMB reconstruction, which gives a mean SMB and SMB trend estimates of 338 +/- 68 GT/yr and respectively −7.3 +/- 2 GT/yr² over 1980-2012.

## 7. Conclusion

This paper describes the methodology and results of the GrIS SMB Model Intercomparison Project (GrSMBMIP): a novel effort that intercompares GrIS SMB fields produced using 5 RCMs, 4 EBMs, 2 PDDs, and 2 GCMs. Model evaluation using ice core data highlights that polar RCMs (in particular MAR and RACMO2.3) have the most accurate representation of SMB in both the GrIS accumulation and ablation zones, but they are also the only ones to have been calibrated to simulate snowfall and melt individually. Biases of other models are nevertheless of the same order of magnitude to those of polar RCMs, which are often used to calibrate these more simple but faster models. The ensemble mean of the 13 inter-compared models compares the best with in-situ SMB observations and is among the best to represent the GRACE-derived mass loss trend between 2003 and 2012. Our results reveal that the mean GrIS SMB of all 13 models has been positive

between 1980 and 2012 with an average of 338 +/- 68 Gt/yr, but has decreased at an average rate of -7.3 +/- 2 Gt/yr[2], mainly driven by an increase of 8.0 +/- 2 Gt/yr[2] in meltwater runoff. The uncertainty has been evaluated to be three quarters of the model spread around the ensemble mean with respect to the in-situ SMB measurements. The good performance of the PDD models in the ablation zone suggests that estimating melt from temperature remains valid under current climate conditions and that the use of more sophisticated energy balance melt schemes can generate larger biases, despite better a priori physics. Finally, the mean (runoff) bias in the ablation zone mostly explains the large discrepancy between models and GRACE-derived mass loss trend in 2003-2012. Moreover, meltwater runoff biases that operate under current climate could strongly impact the models' ability to simulate future melt acceleration as the present day runoff bias should increase in absolute value in the same proportion than runoff under warmer climates, independently of the physics used in the models. This suggests for example that a model overestimating the runoff during the extreme 2012 summer by a factor two should a priori overestimate the future sea level rise coming from the Greenland ice sheet by roughly the same amount, as future SMB changes will mainly be driven by the surface melt increase (Fettweis et al., 2013a).

RCMs have the advantage that they resolve near-surface climate and dynamically downscale the precipitation to higher spatial resolution with respect to their forcing, while PDD and EBM models are fully driven by the near-surface climate of their low resolution forcing fields. Although the two GCMs used in this study are not (or only weakly) forced by historical weather, they simulate the melt reasonably well, which can probably be ascribed to subgrid elevation corrections applied in MPI-ESM and CESM2. However, for precipitation, the native GCM resolution remains too coarse to resolve the spatial variability simulated by RCMs. The spatial variability of precipitation in RCMs is particularly high along the southeast coast of Greenland. However, the paucity of observations prevents us from confirming whether the local high precipitation rates simulated by RCMs, and not captured by lower resolution models, are realistic. Finally, while RCMs are useful tools for evaluating the melt-elevation feedback since they explicitly compute precipitation and melt changes at high resolution on a different ice sheet topography when coupled with an ice sheet model (Le clec'h et al., 2019), running RCMs at a high spatial resolution becomes computationally expensive on time scales beyond one century. This suggests that the PDD/EBM and GCM based approaches may be more suitable for questions that require long simulations (where a coupling with an ice sheet model may be desirable as well), if they well simulate the current climate (in particular melt runoff).

*Author contributions.* XF and SH prepared the manuscript. UKK provides Figure 7 and figures in Supplementary. All authors commented and improved the manuscript.

*Data availability.* All the modelled data sets presented in this study are available from the authors upon request and

without conditions.

*Competing interests.* The authors declare no competing interests.

Acknowledgements. Xavier Fettweis is a Research Associate from the Fonds de la Recherche Scientifique de Belgique (F.R.S.-FNRS). This work was also supported by F.R.S.-FNRS and the Fonds Wetenschappelijk Onderzoek-Vlaanderen (FWO) under the EOS Project n° O0100718F. Computational resources used to perform MAR simulations have been provided by the Consortium des Équipements de Calcul Intensif (CÉCI), funded by the Fonds de la Recherche Scientifique de Belgique (F.R.S.FNRS) under grant 2.5020.11 and the Tier-1 supercomputer (Zenobe) of the Fédération Wallonie Bruxelles
infrastructure funded by the Walloon Region under grant agreement 1117545. Andreas Born and Tobias Zolles acknowledge financial support from the Trond Mohn Foundation. Constantijn J. Berends, Leo van Kampenhout, and Heiko Goelzer have received funding from the programme of the Netherlands Earth System Science Centre (NESSC), financially supported by the Dutch Ministry of Education, Culture and Science (OCW) under grant no. 024.002.001. Edward Hanna acknowledges support from the University of Sheffield's Iceberg high-performance computing team, especially Mike
Griffiths. Philippe Huybrechts acknowledges support from the iceMOD project funded by the Research Foundation – Flanders (FWO-Vlaanderen). Marie-Luise Kapsch and Florian Ziemen were funded by the German Federal Ministry of Education and Research (BMBF) through the PalMod project under grant no. 01LP1504C and 01LP1502A. Michalea King acknowledges support from the National Aeronautics and Space Administration (grant no. 80NSSC18K1027 and NNX13AI21A). Part of the funding for Ruth H. Mottram and Peter L. Langen is provided by the Danish State through the
National Centre for Climate Research (NCKF). Bert Wouters was funded by NWO VIDI grant 016.Vidi.171.063. Brice Noël was funded by the NWO VENI grant VI.Veni.192.019. Masashi Niwano, Akihiro Hashimoto, Teruo Aoki, and Koji Fujita were supported in part by (1) the Japan Society for the Promotion of Science through Grants-in-Aid for Scientific Research numbers JP16H01772, JP15H01733, JP17K12817, JP17KK0017, JP18H03363, and JP18H05054; (2) the Ministry of the Environment of Japan through the Experimental Research Fund for Global Environmental Research Coordination System.
Finally, we would like to thank the Ice Sheet Mass Balance and Sea Level (ISMASS) group, funded by CliC (Climate and Cryosphere), for sponsoring this study and Nanna Karlsson (from Geological Survey of Denmark and Greenland, Denmark) for providing airborne radar transect measurements.

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

| Name | Type | Native Resolution | ERA-Int. Forcing | Downscaling method | Resolution | Main country |
|------|------|-------------------|------------------|--------------------|------------|--------------|
| BESSI | EBM | 10km | yes | | 10km | Norway |
| BOX13 | RCM | ~10km | yes | interpolation | 5km | Denmark |
| CESM | GCM | ~100km | no | elevation classes | 1km | the Netherlands |
| dEBM | EBM | 1km | yes | | 1km | Germany |
| HIRHAM | RCM | 5.5km | yes | | 5.5km | Denmark |
| IMAU-ITM | EBM | 5km | yes | | 5km | the Netherlands |
| MAR | RCM | 15km | yes | | 15km | Belgium |
| MPI-ESM | GCM | ~100km | no | EBM | 1km | Germany |
| NHM-SMAP | RCM | 5km | Yes (JRA-55) | | 5km | Japan |
| PDD1km | PDD | 1km | yes | | 1km | UK – Belgium |
| PDD5km | PDD | 5km | yes | | 5km | UK – Belgium |
| RACMO | RCM | 5km | yes | Statistical downscaling | 1km | the Netherlands |
| SNOWMODEL | EBM | 5km | yes | | 5km | USA |

**Table 1: List of the 13 models used in this study listing the type (Energy Balance Model (EBM), Regional Climate Model (RCM), General Circulation Model (GCM), Positive Degree Day (PDD) model), the forcing (the ERA-Interim or JRA-55 reanalysis), the**
**native resolution, the method used eventually afterwards to downscale the Surface Mass balance (SMB) at 1-km (when it is different than the one use to interpolate all the outputs to 1-km, see Section 2.2) and the country where main development takes place.**

|          | SMB  |         |       | Snowfall |         |       | Runoff |         |       |
|----------|------|---------|-------|----------|---------|-------|--------|---------|-------|
|          | Mean | Std dev | Trend | Mean     | Std dev | Trend | Mean   | Std dev | Trend |
| BESSI    | 387  | 80      | -4.1  | 566      | 54      | 0.3   | 134    | 52      | 4.2   |
| BOX13    | 426  | 99      | -6.5  | 718      | 61      | -0.3  | 508    | 118     | 9.1   |
| CESM     | 421  | 87      | -3.1  | 668      | 59      | 0.1   | 276    | 66      | 4.0   |
| dEBM     | 359  | 121     | -8.1  | 604      | 59      | -0.1  | 280    | 108     | 8.6   |
| HIRHAM   | 398  | 109     | -7.3  | 701      | 63      | -1.5  | 491    | 123     | 8.2   |
| IMAU-ITM | 281  | 129     | -8.7  | 638      | 62      | 0.4   | 382    | 122     | 9.5   |
| MAR      | 372  | 122     | -7.8  | 640      | 55      | -0.5  | 302    | 107     | 8.0   |
| MPI-ESM  | 284  | 101     | -3.5  | 558      | 59      | 0.5   | 336    | 70      | 4.0   |
| NHM-SMAP | 429  | 99      | -4.3  | 807      | 81      | 1.3   | 260    | 79      | 6.1   |
| PDD1km   | 332  | 101     | -6.3  | 519      | 55      | 0.2   | 230    | 87      | 7.0   |
| PDD5km   | 285  | 111     | -6.8  | 534      | 56      | 0.3   | 278    | 97      | 7.5   |
| RACMO    | 357  | 115     | -7.2  | 667      | 59      | -0.7  | 306    | 90      | 6.7   |
| SNOWMODEL | 96  | 179     | -12.9 | 665      | 65      | 0.3   | 469    | 171     | 13.4  |
| **ENSEMBLE** | 338 | 111 | -7.3 | **642**  | **59**  | **0.0** | **331** | **102** | **8.0** |

**Table 2: Mean, interannual variability (standard deviation of the annual means) and linear trend of the main ice sheet SMB, snowfall and runoff in (Gt/yr) over 1980-2012 simulated by the 13 models.**

| | ice cores (# 260; 0.33±0.08mWE) | | | Air-borne Radar (#9043; 0.17±0.13mWE) | | |
|---|---|---|---|---|---|---|
| | Bias | RMSE | Correlation | Bias | RMSE | Correlation |
| BESSI (EBM) | -0.07 | 0.11 | 0.87 | -0.08 | 0.08 | 0.94 |
| BOX13 (RCM) | 0.10 | 0.19 | 0.87 | -0.04 | 0.04 | 0.94 |
| CESM (GCM) | 0.05 | 0.14 | 0.80 | 0.02 | 0.02 | 0.90 |
| dEBM (EBM) | -0.01 | 0.08 | 0.90 | -0.04 | 0.04 | 0.96 |
| HIRHAM (RCM) | 0.02 | 0.13 | 0.83 | -0.01 | 0.02 | 0.96 |
| IMAU-ITM (EBM) | 0.00 | 0.10 | 0.88 | -0.06 | 0.06 | 0.94 |
| MAR (RCM) | 0.01 | 0.08 | 0.93 | -0.01 | 0.02 | 0.99 |
| MPI-ESM (GCM) | 0.01 | 0.12 | 0.76 | 0.00 | 0.04 | 0.82 |
| NHM-SMAP (RCM) | 0.01 | 0.09 | 0.93 | -0.02 | 0.02 | 0.96 |
| PDD1km | -0.01 | 0.04 | 0.97 | 0.00 | 0.01 | 0.96 |
| PDD5km | -0.01 | 0.04 | 0.96 | 0.00 | 0.01 | 0.96 |
| RACMO (RCM) | -0.02 | 0.08 | 0.88 | -0.01 | 0.01 | 0.96 |
| SNOWMODEL (EBM) | -0.05 | 0.12 | 0.87 | -0.09 | 0.09 | 0.95 |
| **ENSEMBLE** | **0.00** | **0.06** | **0.95** | **-0.03** | **0.03** | **0.98** |
| | Snow pits (#130; 0.41±0.34mWE) | | | PROMICE – main ice sheet (#1438; -0.92±0.62mWE) | | |
| BESSI (EBM) | -0.11 | 0.38 | 0.72 | 0.45 | 0.89 | 0.81 |
| BOX13 (RCM) | -0.03 | 0.33 | 0.73 | 0.23 | 0.86 | 0.78 |
| CESM (GCM) | -0.16 | 0.41 | 0.67 | 0.11 | 0.61 | 0.89 |
| dEBM (EBM) | -0.12 | 0.44 | 0.43 | -0.03 | 0.66 | 0.87 |
| HIRHAM (RCM) | -0.10 | 0.38 | 0.66 | 0.09 | 0.57 | 0.91 |
| IMAU-ITM (EBM) | 0.12 | 0.53 | 0.16 | 0.03 | 0.58 | 0.89 |
| MAR (RCM) | -0.08 | 0.37 | 0.68 | 0.10 | 0.48 | 0.93 |
| MPI-ESM (GCM) | -0.08 | 0.35 | 0.75 | -0.05 | 0.69 | 0.85 |
| NHM-SMAP (RCM) | -0.10 | 0.30 | 0.81 | 0.39 | 0.78 | 0.88 |
| PDD1km | -0.09 | 0.44 | 0.40 | -0.18 | 0.69 | 0.89 |
| PDD5km | -0.09 | 0.46 | 0.27 | -0.17 | 0.72 | 0.86 |
| RACMO (RCM) | -0.12 | 0.36 | 0.78 | -0.05 | 0.63 | 0.90 |
| SNOWMODEL (EBM) | -0.17 | 0.47 | 0.33 | -0.32 | 0.61 | 0.92 |
| **ENSEMBLE** | **-0.09** | **0.39** | **0.60** | **0.05** | **0.46** | **0.94** |

**Table 3: Statistics (in mWE) of models vs SMB databases described in Section 3. The number of measurements as well as the mean value and the standard deviation around this mean value for each data set is listed in the titles of the table. Finally, except for both PDD models, the statistics of models vs the SMB PROMICE data base over the main ice sheet or the peripheral ice caps from the common ice sheet mask are given in supplementary material in Table S1.**


|  | Basin 1 | | | Basin 2 | | | Basin 3 | | | Basin 4 | | |
|---|---|---|---|---|---|---|---|---|---|---|---|---|
|  | RMSE | Corr. | Trend | RMSE | Corr. | Trend | RMSE | Corr. | Trend | RMSE | Corr. | Trend |
| BESSI | 10.4 | 0.95 | 3.1 | 18.9 | 0.87 | -4.9 | 20.4 | 0.97 | -35.4 | 32.6 | 0.98 | -62.8 |
| BOX13 | 10.7 | 0.94 | 7.4 | 18.9 | 0.87 | 13 | 22.1 | 0.96 | -35.8 | 32.1 | 0.98 | -88.6 |
| CESM | 13.7 | 0.91 | -5.4 | 23.6 | 0.79 | -31.2 | 44 | 0.87 | -38.8 | 41.2 | 0.97 | -57.7 |
| dEBM | 10.7 | 0.94 | 13.1 | 17.5 | 0.88 | 10.8 | 18.5 | 0.97 | -31.1 | 33.9 | 0.98 | -36.1 |
| HIRHAM | 8.8 | 0.96 | -7.4 | 17.7 | 0.89 | 4 | 15.6 | 0.98 | -24.4 | 24.5 | 0.98 | -48.9 |
| IMAU-ITM | 18.2 | 0.86 | 39.1 | 21.6 | 0.86 | 38.1 | 14.8 | 0.98 | -29.6 | 28.8 | 0.98 | -19.9 |
| MAR | 9.3 | 0.96 | 12.3 | 16.5 | 0.9 | 4.7 | 16.4 | 0.98 | -40.1 | 26 | 0.98 | -39.3 |
| MPI-ESM | 15.9 | 0.88 | -9.2 | 30.7 | 0.64 | -21 | 27 | 0.95 | -16.3 | 56.4 | 0.94 | 4.5 |
| NHM-SMAP | 9.5 | 0.95 | 6 | 18.5 | 0.87 | 3.6 | 15.8 | 0.98 | -19.1 | 25.5 | 0.98 | -75.3 |
| PDD1km | 11.9 | 0.93 | -5.8 | 17.6 | 0.88 | -14.6 | 19.1 | 0.97 | -12.7 | 30.5 | 0.98 | 0 |
| PDD5km | 11.8 | 0.93 | 8.5 | 15.2 | 0.91 | 2.3 | 19.6 | 0.97 | -14.2 | 30.4 | 0.98 | -5.3 |
| RACMO | 9.8 | 0.95 | 12 | 17.3 | 0.89 | 12.8 | 16.3 | 0.98 | -18.2 | 26.3 | 0.98 | -38.9 |
| SNOWMODEL | 18.7 | 0.86 | 66 | 27.2 | 0.81 | 89.7 | 13.8 | 0.98 | -8.4 | 23.8 | 0.99 | -17.8 |
| **ENSEMBLE** | **10.5** | **0.95** | **10.5** | **16.7** | **0.89** | **12.3** | **16.8** | **0.98** | **-23.5** | **27.7** | **0.98** | **-40.3** |
|  | Basin 5 | | | Basin 6 | | | Ice sheet | | |  |  |  |
|  | RMSE | Corr. | Trend | RMSE | Corr. | Trend | RMSE | Corr. | Trend |  |  |  |
| BESSI | 45.2 | 0.94 | -41.4 | 17 | 0.99 | 16.2 | 95.5 | 0.98 | -113.7 |  |  |  |
| BOX13 | 39.4 | 0.96 | 2.7 | 15 | 0.99 | -14.5 | 84 | 0.99 | -103.1 |  |  |  |
| CESM | 51.4 | 0.93 | -41.4 | 21.5 | 0.98 | -8.9 | 95.7 | 0.98 | -165.9 |  |  |  |
| dEBM | 43.5 | 0.95 | 0.1 | 19.1 | 0.99 | 20.6 | 94.5 | 0.98 | -9.9 |  |  |  |
| HIRHAM | 26 | 0.98 | -1.1 | 14.5 | 0.99 | -21.5 | 54.2 | 0.99 | -87.3 |  |  |  |
| IMAU-ITM | 43.7 | 0.95 | 2.7 | 26.5 | 0.98 | 30.7 | 101.2 | 0.98 | 71.6 |  |  |  |
| MAR | 26.8 | 0.98 | 6.7 | 14.2 | 0.99 | -1 | 51.7 | 0.99 | -42.4 |  |  |  |
| MPI-ESM | 90.3 | 0.8 | 31 | 39.9 | 0.96 | -29.8 | 193.6 | 0.94 | -28.3 |  |  |  |
| NHM-SMAP | 32.7 | 0.97 | -52.6 | 17 | 0.99 | -29.7 | 71.3 | 0.99 | -153.4 |  |  |  |
| PDD1km | 42.7 | 0.95 | -19.9 | 18.7 | 0.99 | 11.4 | 93.5 | 0.98 | -28 |  |  |  |
| PDD5km | 40.6 | 0.95 | -6.4 | 19.6 | 0.99 | 27.7 | 83.5 | 0.99 | 26.1 |  |  |  |
| RACMO | 26.9 | 0.98 | -4.5 | 14.6 | 0.99 | -9.4 | 56.7 | 0.99 | -34.3 |  |  |  |
| SNOWMODEL | 43.3 | 0.95 | 85.8 | 23.7 | 0.98 | 79.2 | 93.3 | 0.98 | 304.2 |  |  |  |
| **ENSEMBLE** | **35.1** | **0.96** | **-3.6** | **16.2** | **0.99** | **9.6** | **73.2** | **0.99** | **-22.3** |  |  |  |

**Table 4: Statistics of models (in which the signal coming from the ice discharge has been subtracted from SMB) vs GRACE for each basin and the whole ice sheet. The trend (in Gt/year) shows the linear trend that must be applied to match GRACE estimates. So negative numbers mean that the downward trend is underestimated compared to GRACE. RMSE (root-mean-square error) and correlation are computed after having applied this trend to the modelled time series.**


| | bare ice area and SMB > STD/2 | bare ice area and SMB < - STD/2 | Agreement |
|---|---|---|---|
| BESSI | 3.5 | 1.1 | 95.5 |
| BOX13 | 0.9 | 5.0 | 94.0 |
| CESM | 1.9 | 1.6 | 96.4 |
| dEBM | 0.6 | 2.9 | 96.4 |
| HIRHAM | 0.6 | 2.4 | 97.0 |
| IMAU-ITM | 0.5 | 9.2 | 90.4 |
| MAR | 0.4 | 3.6 | 96.0 |
| MPI-ESM | 0.8 | 2.3 | 96.9 |
| NHM-SMAP | 0.6 | 3.7 | 95.8 |
| PDD1km | 1.4 | 2.3 | 96.3 |
| PDD5km | 1.0 | 5.3 | 93.7 |
| RACMO | 0.7 | 3.0 | 96.3 |
| SNOWMODEL | 0.3 | 14.1 | 85.6 |
| **ENSEMBLE** | **0.4** | **4.2** | **95.3** |

**Table 5: left) Percentage of the main ice sheet area where presence of bare ice area detected in MODIS on average over 2000-2012 and where the modelled mean SMB is significantly positive. The half of the interannual variability is used to evaluate the statistical significance of the equilibrium line. Middle) the same for (no) presence of bare ice and where SMB is significantly negative. Right)**
**Percentage of agreement between modelled ablation zone and bare ice area from MODIS.**

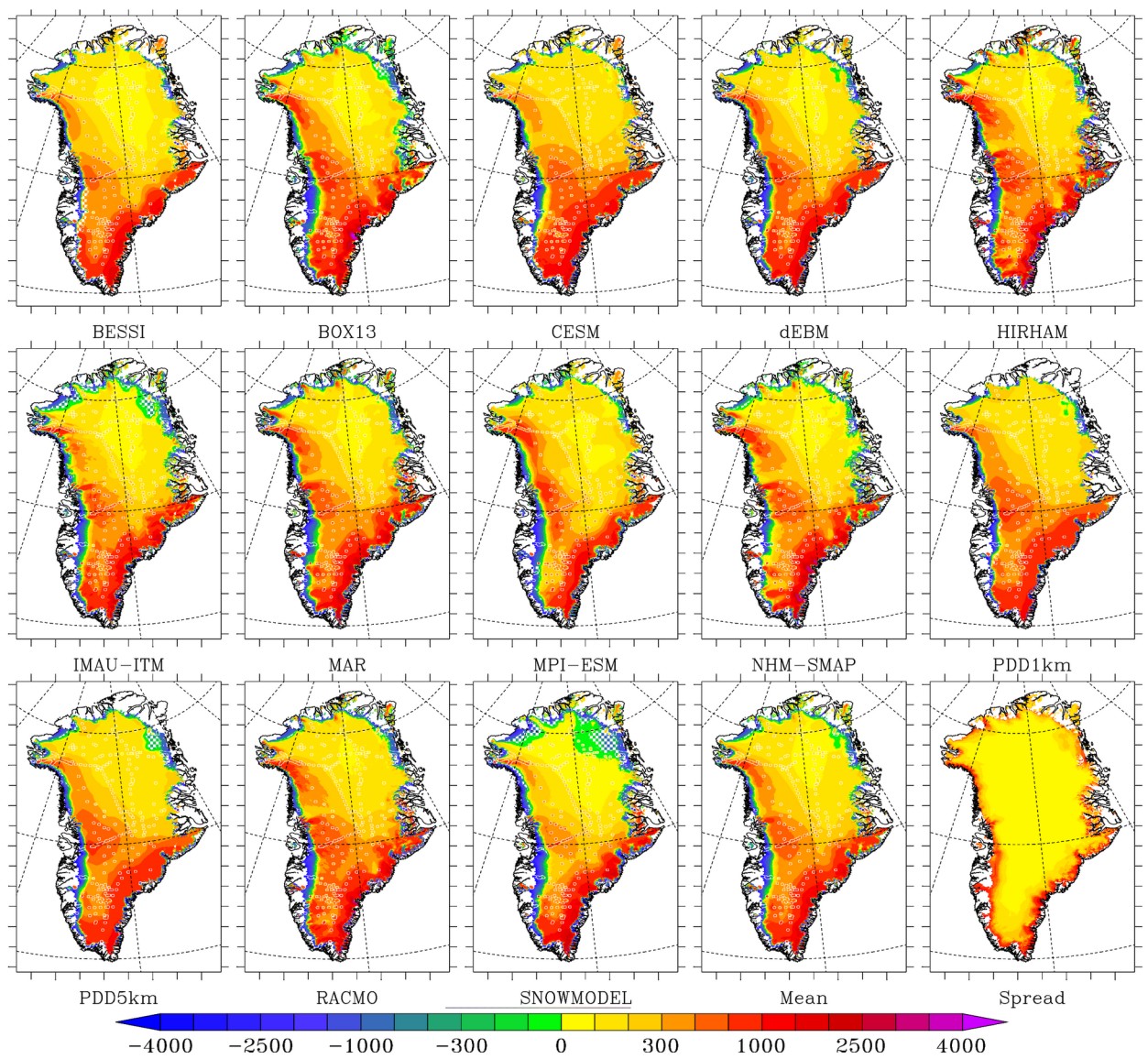

**Fig. 1: Mean SMB (in mmWE/yr) over 1980-2012 simulated by the 13 models as well as the ensemble model mean and the spread around this mean. The SMB measurements (ice cores + PROMICE) used to evaluate the models are represented as white circles. The areas listed in Table 4 where SMB disagree with the satellite derived bare ice area are shown in hatch. Finally, it is important to note that for better visibility, the scale is not linear by using a step of 100 for absolute values lower than 500 mmWE/yr and a step of 500 above.**


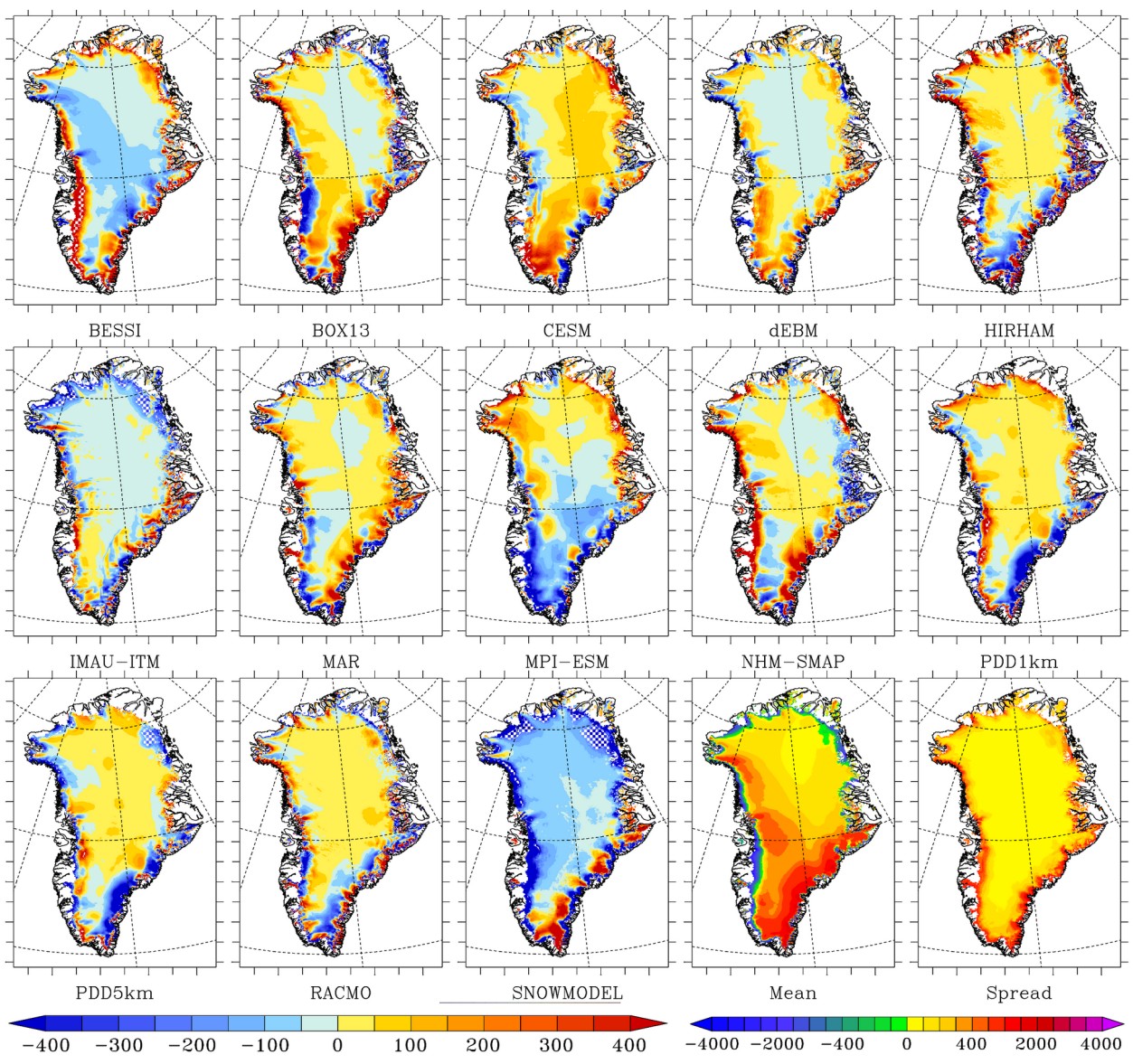

**Fig. 2: Same as Fig 1 but for the modelled SMB vs the ensemble model mean over 1980-2012 (shown in the 2 last plots). As Fig. 1, the areas listed in Table 4 where SMB disagrees with the MODIS-derived bare ice area are shown in hatch.**


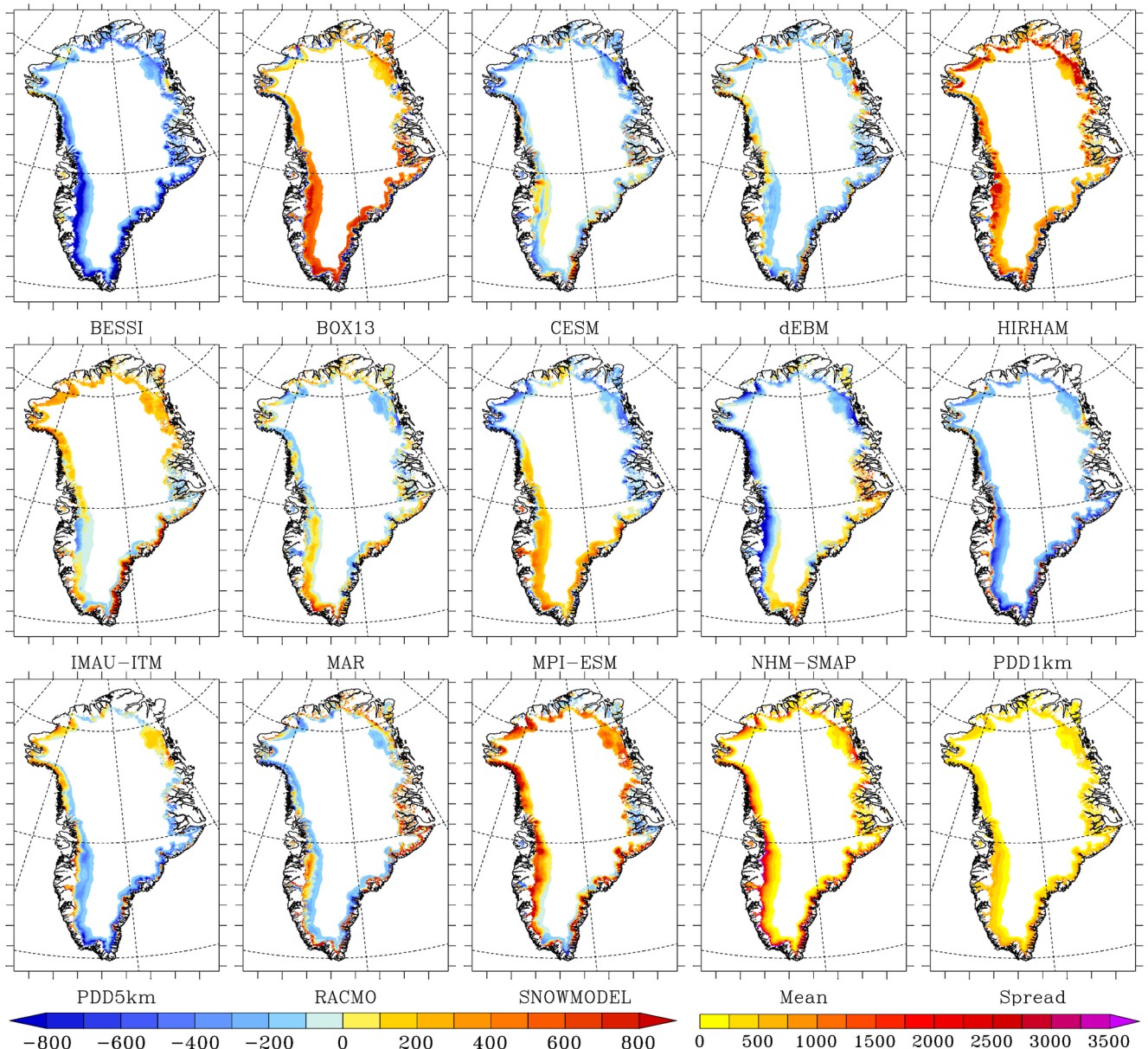

**Fig. 3: Same as Fig 2 but for the modelled runoff. The ensemble mean and models spread around the mean are also shown in mmWE/yr in the two last plots. Finally it is important to note that only the area where the runoff of the ensemble mean is higher than 100mmWE/yr is shown here.**

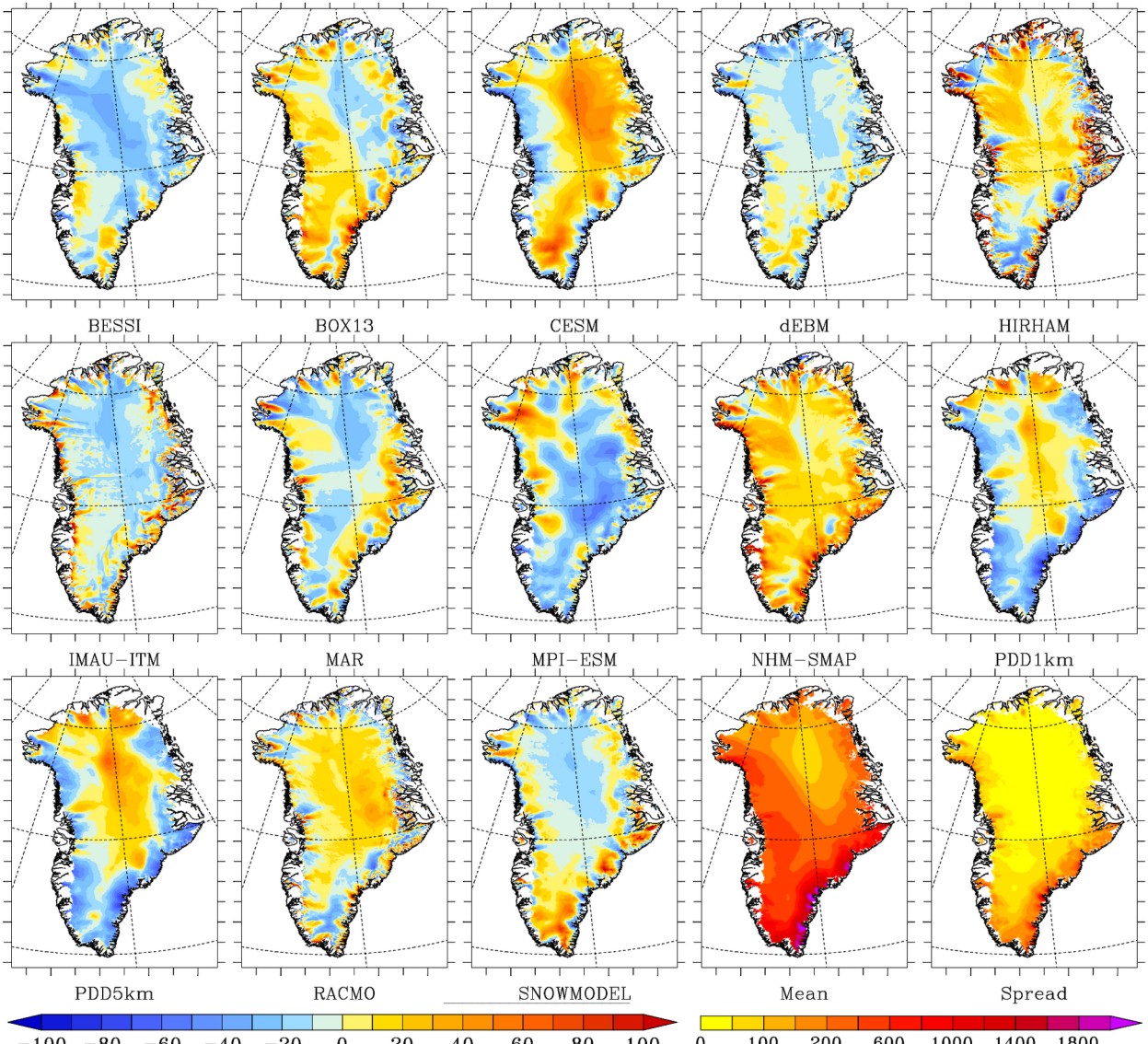

**Fig. 4: Same as Fig 2 but for the modelled snowfall (linearly interpolated on the 1 km common grid but without any elevation correction) vs the ensemble model mean over 1980-2012 as a percentage of the ensemble model mean of snowfall accumulation. The ensemble mean and models spread around the mean are also shown in mmWE/yr in the two last plots.**

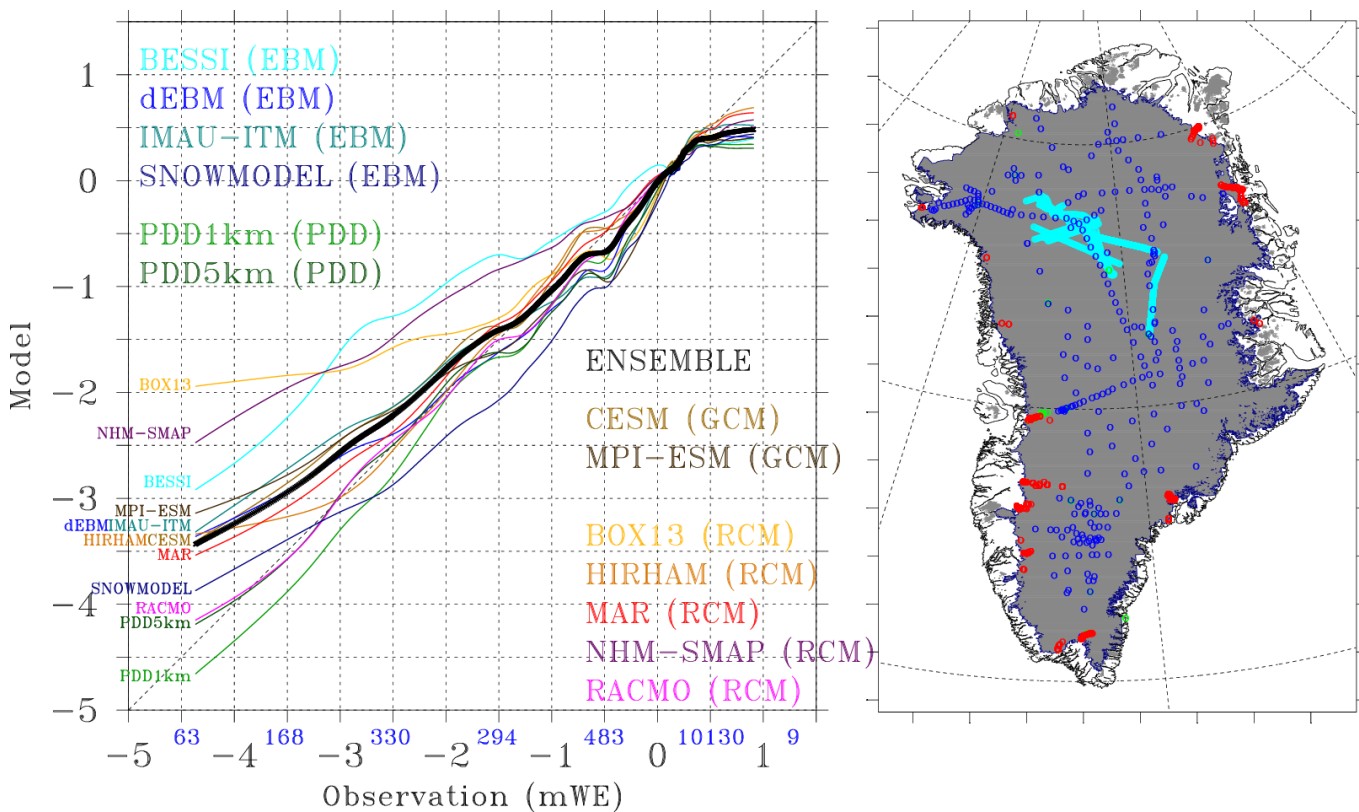

**Fig. 5: Left)** Scatter plot of modelled vs measured SMB in mWE. To increase the visibility of this figure, a running mean on 200 samples has been applied here after having sorted the samples (observation, model) on the observations. The numbers in blue on the X-axis indicate the number of observations with SMB values within each interval of the X-axis. **Right)** Locations of the in-situ measurements: ice cores in dark blue, airborne radar transects in light blue, snow pits in green and PROMICE in red.

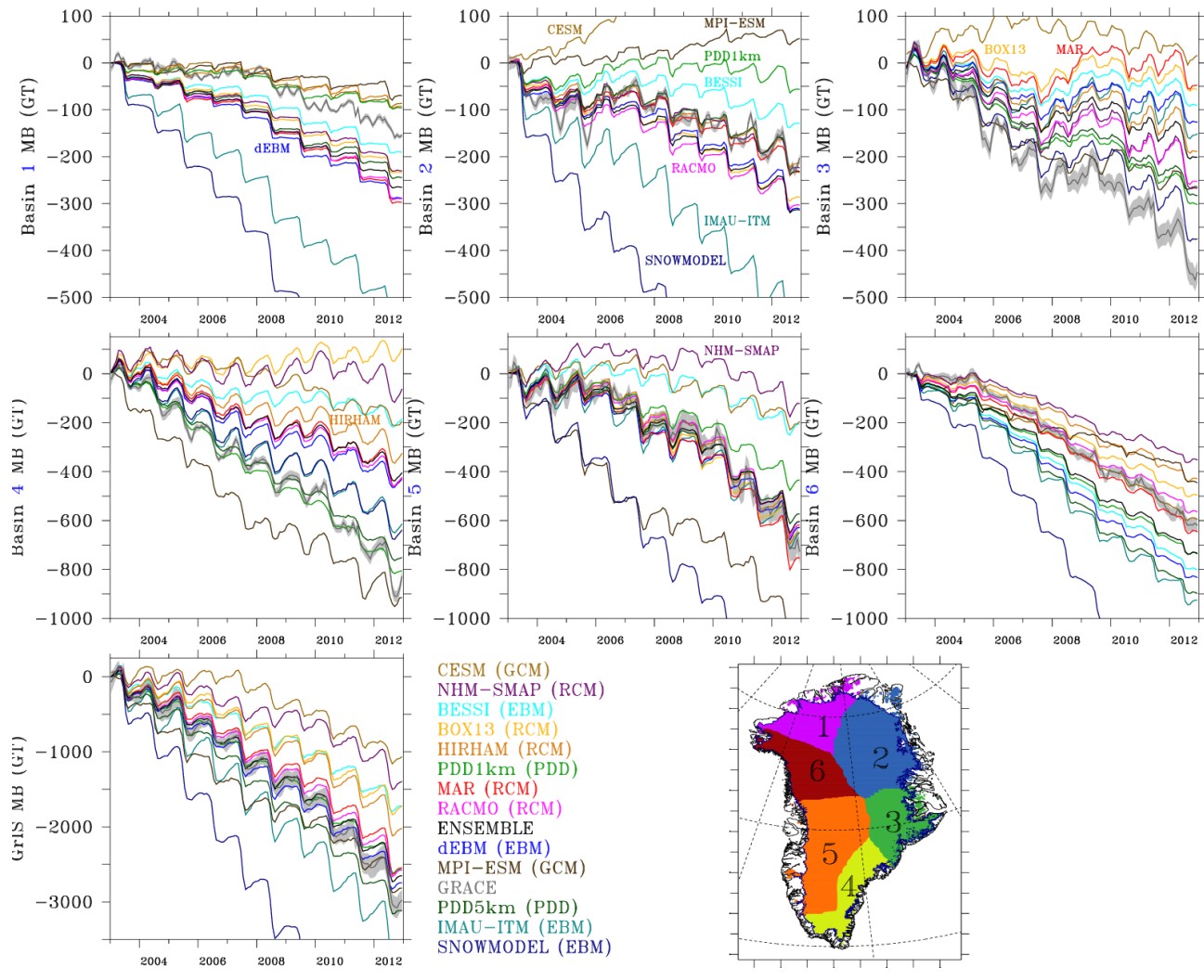

**Fig. 6: Time series of the mass balance (MB) changes from GRACE over 2003-2013 as well as from the 13 models in which the signal from ice discharge (King et al., 2018) has been subtracted from the modelled SMB. Times series are shown for 6 basins as well as over the whole ice sheet. Except for both PDD models, the ice caps from the common ice sheet mask are included and the tundra areas are discarded. Finally, the legend is sorted in order of the mass balance changes estimated over the whole ice sheet.**

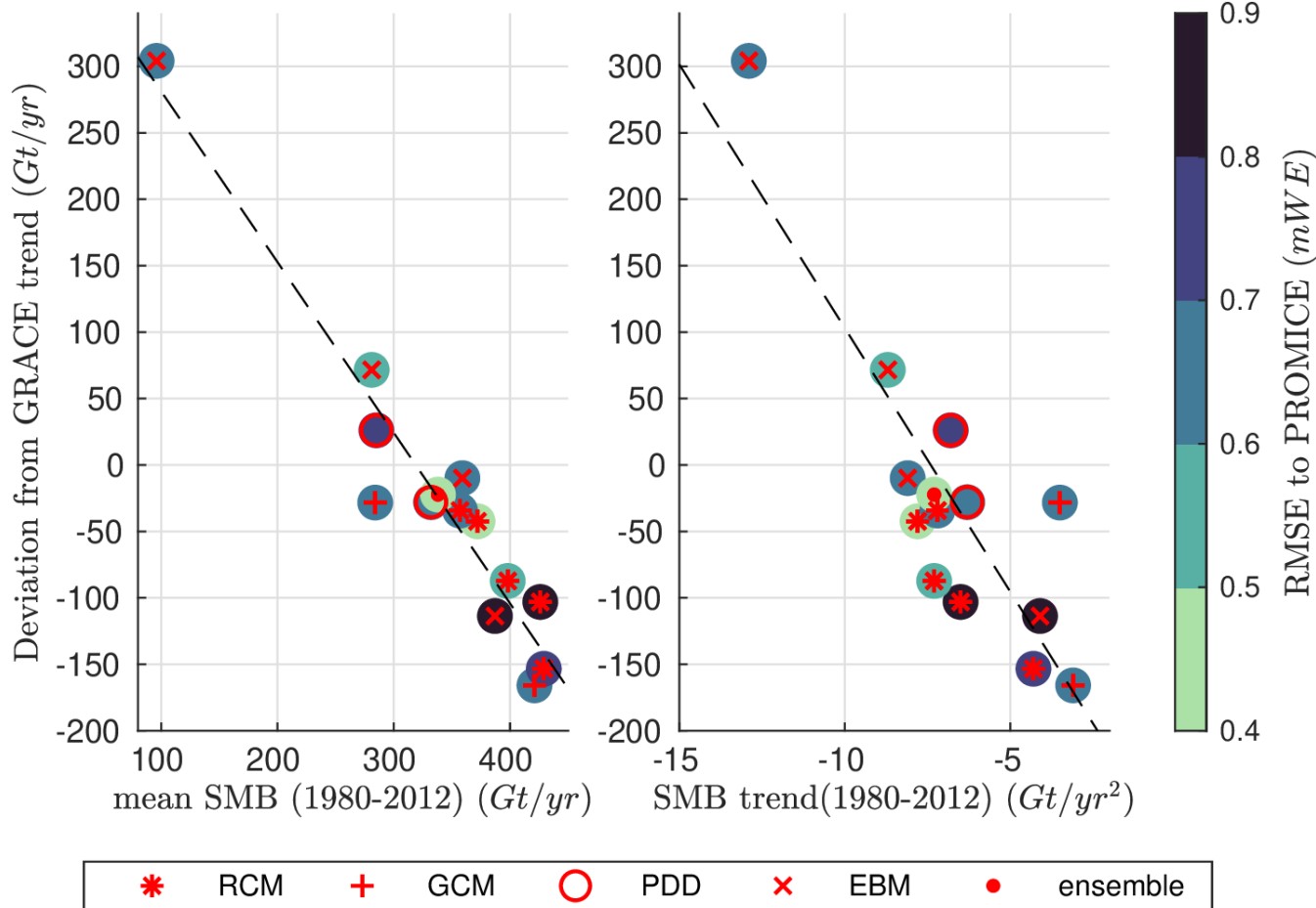

**Fig. 7: Left)** Scatter plots of the deviation from GRACE trend (in GT/year) vs mean SMB over 1980-2012 (in GT/yr), listed respectively in Table 4 and Table 2. **Right)** Same as left but with the SMB trend over 1980-2012 (in GT/yr²). On both plots, the colour background gives the RMSE with the PROMICE SMB data base (see Table 3).