# Peer review of "GrSMBMIP: Intercomparison of the modelled 1980-2012 surface"

_The Cryosphere, 2019_

## Referee Comment (RC1) · Anonymous Referee #1 · 28 Feb 2020

The following is a review of "GrSMBMIP: Intercomparison of the modelled 1980-2012 surface mass balance over the Greenland Ice sheet" By X. Fettweis, et al.

The manuscript presented describes the Greenland Ice Sheet surface mass balance (SMB) model inter-comparison results for the historical period 1980-2012. The authors assess the ability of different types models (including regional climate models, radiation balance models, positive degree day models, and general circulation models), 13 in all, to estimate the surface mass balance over the Greenland continent. Skill criteria for the models are derived from observational datasets, including MODIS bare ice extent, ice cores/snow pits/ in-situ observations, and the calculation of regional SMB as the

difference between GRACE estimates of mass and previously-published ice discharge into the ocean. A large amount of effort is taken to design the experiment and compile model submissions that have the same source of forcing, the same spatial resolution, and cover the same overlapping time periods. Through this comparison, the authors derive an ensemble mean and standard deviation of Greenland SMB over the 1980-2012 period, as well as trends. The authors find the largest model discrepancies are along the ice sheet margins, and it is the increase in meltwater runoff along the margins that drive the prevailing negative trend in Greenland mass balance over the study period. Results suggest that regional climate models have strong skill in matching observed patterns of SMB, though computationally expensive compared to the positive degree day or radiation balance models. Overall, the authors find that it is the ensemble mean that best matches observations, meaning that errors from the various models balance each other out and do not convey any obvious systematic biases.

The work presented here is critical for cryosphere scientists, especially to the scientists interested in quantifying and simulating the evolution of ice sheets (including atmosphere/surface/ice sheet modelers). This is clearly a massive effort, and as observation of SMB in many areas are quite sparse, the authors have done a very nice job of compiling meaningful comparison criteria as a first attempt at this type of exercise. Such an effort is quite necessary to build a SMBMIP community and launch similar efforts in the future. The work presented here is especially a nice basis on which to build future comparison efforts that may focus on sea level projections. This is especially true considering the conclusion that the current compilation of models does not show systematic bias. For these reasons, publication of this work is timely and critical.

That being said, the manuscript in its current form needs a lot of work, especially the text which requires major revision. The tables and figures, in general, are adequate for conveying the discussion and conclusions of the manuscript. However, the model descriptions take up most of the text, and the rest is very concise. I think expanding upon the scientific results would make this manuscript much less of a technical paper

and much more appropriate for publication in The Cryosphere. Such improvements would also help broaden the audience for this paper. As is, the manuscript is difficult to digest by other cryosphere scientists, and the authors do not make it immediate clear to the reader why these impactful results may be of interest to their research.

Below, I outline my general comments to the authors:

Introduction – In general, the introduction should be expanded to discuss more clearly the topics of observed variability in SMB over the historical time period assessed and why it is important and/or difficult to capture them with models. In addition, an introduction to the types of models that are assessed should be given, since those reading this manuscript might not be familiar with how and why these types of models differ. This could be a good way to let the reader know about the general advantages and disadvantages of each model type also. Another helpful topic to cover would be a short discussion on why this effort is so important and what the authors are aiming to learn about model bias (i.e. why a historical assessment is helpful to complete before assessing projections from this group of models). This pertains to statements that are made in the conclusion section of the paper, especially those about implications on model coupling and about quantification of uncertainties in sea level rise projections. Introducing these concepts before mentioning them in the concluding remarks would help highlight their importance and future inter-comparison goals.

Additionally:

- The first paragraph of the introduction mentions glacial water storage, and notes that this is the first time it has been evaluated. However, the GS term is not included in Equation 1, and GS is not discussed explicitly anywhere else in the manuscript. Please be more specific here about how GS is evaluated, and how it is being included in this analysis.

- Line 70-71, This line might be better coming after Eq. 1.

- Line 78, Please specify the type of variability you refer to here

Model Section –

- Maybe it would be helpful if there was an overarching Methods section, since the Model section would really benefit from a short introduction describing what your methods in general are, and what you plan to do as an inter-comparison exercise. If the Model, Observations, and Evaluation all came under a larger section, it might be a good way to add some explanation prior to the reader going through all the details right away without understanding what type of inter-comparison is being presented.

- As it is, it is very difficult and quite boring for the reader to be introduced to a list of models and their descriptions up front with no introduction to them. Maybe a table of model names, types, native resolution, downscaling type, etc., could help serve as a reference/summary to this section. Such a table/figure might help the reader to have something to refer to while looking through the tables and the figures. Easier access to model type (by a table or color coding in the figures?) would help make the results easier to read.

- If at all possible, it also might be helpful to push this list deeper into the section – maybe with the observations or data described first? (Though this might be fixed by section summary I mention earlier).

- It is also important to note, that many model descriptions refer to RACMO within their write-ups, but no reference for RACMO, what it is, or what it stands for has been included prior to these sections.

GRACE estimation Section

- It could be helpful to include the equation of glacier mass balance here, so that it is clear to the reader how SMB is calculated from GRACE and ice discharge.

- These last two sentences can probably be simplified to just say that you are using the methods of King et al. (2018), but instead of RACMO, you use each of your different

[Figure]

SMB products. These sentences are awkward that way that they read currently.

Comparison with GRACE measurements Section – Results here are very interesting and there is plenty to point out to the reader. In general, I don't think there is any advantage to being extremely concise. It would be nice for you to lead the reader from figure to conclusion for some of the statements made in this section.

- The discussion about the seasonal cycle here is a nicely suggested by your presented results. Could you please add some more explanation in order to lead the reader a bit more on why the RMSE from your Supp. Fig. 2 would imply how the seasonal cycle is modeled?

- For discussing the GCM's, could you please be explicit about the difference between the forcing of variability on these models by ERA-Interim, and how it pertains to the RMSE?

Conclusion Section – Here, some of new concepts that were not brought up earlier in the manuscript are mentioned. This includes the mention of coupling with an ice sheet model (i.e. a topographic feedback scheme was not used, maybe add a reference to a paper that shows the feedback may be important) and quantification of uncertainty in climate projections. I think the manuscript would be improved if some space was taken in the discussion section to mention more of how results presented here do have implications for these other applications. Implications of interest could range from estimates of historic sea level contribution to forcing of ice sheet models for historic and future simulations, and the ability to now give those applications error bars. I would even say that bringing these applications up in the Introduction as justification for conducting this MIP could help improve the manuscript's impact.

Below, I offer some more specific comments/suggestions:

Page 2, line 62: Please rephrase, "of the same order as RCMs compared with observations and therefore remain useful tools. . ." or something similar

Page 3, line 98: maybe, "although each model prescribes the reanalysis forcing in different manner". Please refrain from referring to the forcing as data.

Page 3, line 99: "(EMBs)"

Page 11, lines 1-2: A reference to Fig. 1 would be helpful here

Page 11, line 336-337: "This allows. . ." Please rephrase this sentence. It is very awkward and difficult to understand.

Page 12, line 359: maybe, ". . .compared to the resulting mass balance estimates from the GRACE product".

Page 13, line 411: "with the GRACE-derived. . ."

Page 15, line 452: Instead of mainly, maybe "largely"?

Page 15, line 457: Not sure what you mean by "oscillates" in this context. Maybe "deviates from the mean"?

---

## Referee Comment (RC2) · Anonymous Referee #2 · 19 Mar 2020

The manuscript presents an experiment in which the surface mass balance (SMB) output of five regional climate models, four surface energy balance models, and two positive degree day (PDD) schemes for the Greenland Ice Sheet (GrIS) are each forced with ECMWF-Interim atmospheric reanalyses over the period 1980-2012. They are compared with each other, with available in situ observations, with MODIS-derived bare ice extent, and with a derived gravimetric data set in which an observed terminal glacier discharge has been incorporated. The output from two global general circulation models is also considered. The main results presented are that the models simulate a statistically significant decrease in SMB over the period, that the largest differences between models occur on the ice sheet margins, and that regional climate

models generally perform well in comparison to the validation data.

The manuscript is around 7500 words with 6 figures and 4 tables, which is a reasonable length for the topic. It represents a considerable community effort in organizing and executing the experiment. The author list comprises most major modeling efforts for contemporary GrIS SMB with the outstanding exception of atmospheric reanalyses (e.g., the Arctic System Reanalysis; MERRA-2). The initial reaction is that this is a significant update on earlier efforts of Vernon et al. (2013) and perhaps Rae et al. (2012) in model assessment. While those studies were mostly focused on regional climate models, this manuscript aspires beyond that with the inclusion of a large number of surface energy balance and PDD models. I have a few points for the authors to consider below, and so would suggest some revision of the manuscript.

1. The experiment necessarily relies on the common use of one forcing product, ECM-Interim. By itself, this study is then not a complete characterization of SMB and its uncertainty from model sources, as the uncertainty of the forcing would also need to be considered. The use of different forcing products is beyond the scope of this study, but it would seem that the forcing selection plays a significant role in determining trends. Consider that if one wished to comprehensively evaluate a forcing product for the GrIS, a possible approach would be this experiment: a comparison of many forced models with observations may be seen as an elaborate validation of the forcing product. Is that not so? The purpose of this study is an appraisal of the different models, and for this purpose the key results are in how models compare with each other, and the systematic differences between them. These would seem to be the results that should be emphasized. Comparisons with observations are of interest (e.g., Fig. 1 is very interesting) but would not seem to be the principal outcome to be emphasized. The manuscript presents a considerable amount of information on the intercomparison in the form of figures and tables. Beginning in section 4, the text focuses primarily on the direct comparison with observations. The intercomparison is largely covered in the second paragraph of section 5. It is suggested that the results be re-ordered with the

intercomparison presented first. Some additional aspects of the intercomparison may be highlighted, as suggested below.

2. For tables and Figs. 2, 4, 5, and 6, the plots should be sorted by model type rather than alphabetically, and perhaps labelled accordingly. It may also be useful to plot the spread for each model type.

3. It is useful to continually compare this experiment with the previous efforts cited in the introduction. Vernon et al. found SMB estimates were within 34% of the multi-model mean of 4 models. Table 4 suggests this value is now something like 22% for the 5 RCMs but close to 100% when all of the models are considered. Does this suggest an increasing proficiency within the RCMs. Also, it is noticeable that the manuscript does not indicate surface temperature sensitivity. It is difficult to include and assess the PDD and EBM models without that consideration. For example, this was a focus of Bougamont et al., who found that PDD models were more sensitive than EBM models. Given the same forcing and the trends shown in Table 2, it does not appear that a similar conclusion holds here, is that correct?

4. At 445 words, the abstract is too long by half. A large part of the abstract is devoted to motivation, which should instead be mostly left to the introduction. As suggested in the previous point, may consider adding more text regarding the resulting differences between the models.

5. A concern is the very lengthy description of the models contained in section 2. The descriptions include sub-model components, the forcing time scale, vertical resolution, ancillary forcing data, etc. It is of course useful for close examination of individual model results, but this is generally available elsewhere from the cited literature, and it is not clear that all of it is directly pertinent to the aggregate results presented for understanding cryospheric modeling. It is suggested that this material may be incorporated into supplementary text and/or condensed with a table that includes model type, references, and major points. Otherwise it could be argued that this type of material is

more appropriate for a publication such as Geoscientific Model Development.

6. Lines 455 and following. As indicated, it is apparent from Fig. 6 that the snowfall from the EBM models, which is directly imported from the forcing, is low (mostly blue) compared to the mean over the interior regions of the ice sheet. Would it be correct in saying that models that compute snowfall generally show higher amounts than the forcing? This appears to be true for most of the HIRHAM, NHM-SMAP, and RACMO RCMs and to some extent for the BOX13. Is that an expected systematic response?

7. A list of acronyms in the appendix would be useful.

---

## Author Response (AR1)

Dear Editor,
Dear Robin,

Please find a revised version of our manuscript.

According to the reviewers recommendations and as proposed in our responses to reviewers (see the discussion in TCD), we have

1. shorted our abstract as requested.

2. improved our introduction to put better in the context the interest of such a models intercomparison as well as to better describe the advantages and drawbacks of each kind of models.

3. added a table (Table 1) before the description of the models (in Section 2) to summarise the information given afterwards (model name, type, resolution, forcing, …)

4. started the discussion with the model intercommunion (Section 4) for ending with the evaluation with observations (Section 5). As we have inverted these section, the text is highlighted as new in the track change version while the main text is basically the same than before, except some minor improvements.

5. discussed more in-depth (see Section 5.2) the comparison with GRACE by separating the evaluation of modelled seasonal cycle (accumulation in winter and melt in summer) and interannual variability in this seasonal cycle to the long climate variability (linked in part to global warming) induced mass loss.

6. added a new Section 6 (Discussion) and a scatter plot (Fig7) comparing the bias with GRACE against the mean SMB simulated by the models. This allowed us to propose a best estimate of the current SMB based on the ensemble mean and to discuss the uncertainty of this best estimate. This new Section 6 is the main change with respect to the previous version of our manuscript.

8. generally improved the text of our manuscript with respect to the suggestions of the reviewers.

Except these listed changes, the presented model results as well as the statistics in the tables are exactly the same than before.

Thanks for considering this revised version.

Best regards,

Xavier, on the behalf of all the co-authors.

[revised manuscript text omitted]

Abstract. The Greenland Ice Sheet (GrIS) mass loss has been accelerating at a rate of about 20 +/- 10 Gt/yr² since the end of the 1990's, with around 60% of this mass loss directly attributed to enhanced surface meltwater runoff. However, in the climate and glaciology communities, different approaches exist on how to model the different surface mass balance (SMB) components using: (1) complex physically-based climate models which are computationally expensive; (2) intermediate complexity energy balance models; (3) simple and fast positive degree day models which base their inferences on statistical principles and are computationally highly efficient. Additionally, many of these models compute the SMB components based on different spatial and temporal resolutions, with different forcing fields as well as different ice sheet topographies and extents, making inter-comparison difficult. In the GrIS SMB model intercomparison project (GrSMBMIP) we address these issues by forcing each model with the same data (i.e., the ERA-Interim reanalysis) except for two global models for which this forcing is limited to the oceanic conditions, and at the same time by interpolating all modelled results onto a common ice sheet mask at 1 km horizontal resolution for the common period 1980-2012. The SMB outputs from 13 models are then compared over the GrIS to (1) SMB estimates using 
[revised manuscript text omitted]

Additionally, the statistics listed in Table 2 are useful for evaluating the seasonal variability once the linear trend in both time series has been removed. In comparison to GRACE we find that the five RCMs simulate the seasonal cycle of SMB much better than other types of models (see Fig. S2 in supplementary), although the trend (Fig. 3) is significantly underestimated in e.g. HIRHAM and NHM-SMAP. The two GCMs have the larger RMSE mainly because their interannual variability is not (fully) forced by ERA-Interim. Finally, it is interesting to note that, for most of the models, the sign of the bias with the PROMICE data set (see Table 1) is highly correlated to the sign of the trend bias with respect to the GRACE based product. For example, the 2003-2012 changes in SMB were driven by an increase of melt, and those models underestimating surface ablation also underestimate these recent changes (the signal coming from discharge change is the same for all the models). Finally, for a total surface mass loss over 2003-2012 of ~ 3000 Gt as suggested by the GRACE data set, the models range from -1066 Gt to -6034 Gt with an ensemble mean of -2611 +/- 1253 Gt suggesting a large discrepancy between models and therefore a large uncertainty in the modelled SMB trends.

**5.24.3 Comparison with GRACE measurementsbare ice extent**

To enable comparison with GRACE mass change, we estimate total mass balance (MB) for each model in 6 basins (see Fig. 6 and Table 4) by subtracting observed ice discharge D (King et al., 2018) from modelled SMB for the 13 models, as King et al. (2018) originally did using the RACMO-based SMB:

We can reasonably assume that the mean SMB should be negative in the bare ice area and positive above the snow line. However, the equilibrium line altitude varies each year. Therefore, we have chosen to only use SMB values that fall within 0 mmWE/yr plus (resp. minus) half the SMB interannual variability (/2) to detect the modelled accumulation (resp. ablation) zone. Except BESSI, all models are able to develop a large enough bare ice area, although most of them overestimate the ablation zone extent, in particular IMAU-ITM and SnowModel (see Table 3). In Fig. 2, the hatched areas outline the regions where the models overestimate or underestimate (only for BESSI) the bare ice area. We can see that BESSI fails to represent the extent of the south-western ablation zone. Conversely, IMAU-ITM, BOX13, PDD5km and SnowModel overestimate the extent of the ablation area in north-east Greenland, where the SMB from the other models is

also very low but remains positive. Finally, it is interesting to note that both GCMs (CESM2 and MPI-ESM), despite their coarse native resolution in the atmosphere, are able to accurately model the snow line, which is attributed to their subgrid downscaling module.

**5. Discussion and comparison**

Integrated over the common main ice sheet mask (see Table 4), the average total GrIS SMB over 1980-2012 ranges from 96 Gt/yr (SnowModel) to 429 Gt/yr (NHM-SMAP), with a mean value of 340 +/- 112 Gt/yr. Comparing the two largest SMB components (i.e. snowfall and runoff), we show large discrepancies between models. For some models, SMB falls within the range of the other models only due to compensating effects of over or underestimating both snowfall and runoff (see Fig. S3 in supplementary). For example, the snowfall and runoff from BESSI and the PDD models are very low compared to other models but yield similar integrated SMB. In addition, the SMB of NHM-SMAP (resp. SnowModel) is substantially higher (resp. lower) than that of other models, due to an overestimation in snowfall accumulation (resp. meltwater runoff). Finally, except for SnowModel (which suggests an SMB trend close -12.9 Gt/yr$^2$), all models suggest that the SMB of the GrIS has decreased at a rate of ~7 Gt/yr$^2$ over the period 1980-2012, primarily due to an increase of meltwater runoff (~ +8 Gt/yr$^2$).

$$MB = SMB - D \hspace{4cm} (2)$$

If we compare each model to the ensemble mean (Fig. 4), we can see that BESSI, NHM-SMAP and PDD1km generally simulate lower runoff in the ablation zone with respect to the other models (Fig. 5). In contrast SnowModel, HIRHAM and BOX13 simulate rather larger runoff than the ensemble mean. These differences mainly explain the SMB anomaly in the ablation zone shown in Fig 4 with respect to the ensemble mean. IMAU-ITM, BESSI and SnowModel are too dry in the interior of the ice sheet, even though they use identical ERA-Interim precipitation forcing as the other PDD/EBM models (Fig. 6). In the accumulation zone of south Greenland, CESM overestimates snowfall rates while MPI-ESM underestimates them in addition of overestimating runoff in this area. Finally, for all the RCMs, the snowfall accumulation does not show similar and systematic anomalies over a large extent and oscillates around the ensemble mean. This a priori better representation of precipitation spatial variability in the RCMs is likely due to the fact that the precipitation is computed at higher resolution than in the GCMs or in ERA-Interim reanalysis used to force the PDD/EBM models. This highlights the advantage of simulating precipitation at high spatial resolution in order to represent the interaction between the atmospheric flow and ice-sheet topography. The south-east coast of Greenland shows the largest discrepancy between models, reaching 2 mWE/yr locally, and is where most RCMs simulate higher precipitation than other types of models. Unfortunately, the data coverage along the south-eastern coast is very sparse, making it hard to prove whether high

650  accumulation rates in RCMs, locally exceeding 3 mWE/yr, are actually realistic. This highlights the need for a higher density of in situ measurements in southeast Greenland where the models simulate the maximum of precipitation. Shallow ice radar or remote sensing (elevation satellite) could also help to evaluate the accumulation rates in this area.

**6. Conclusion**

This paper describes the methodology and results of the SMB Model Intercomparison Project (GrSMBMIP): a novel effort
655  that intercompares GrIS SMB fields produced using 5 RCMs, 4 EBMs, 2 PDDs, and 2 GCMs. Model evaluation using ice core data highlights that polar RCMs (in particular MAR and RACMO2.3) have the most accurate representation of SMB in both the GrIS accumulation and ablation zones but they are also the only ones to have been calibrated to simulate separately snowfall and melt which not all models do. Biases of other models are nevertheless on the same order to those of polar RCMs (which are often used to calibrate these more simple but faster models) and the ensemble mean of the 13 inter-
660  compared models compares best with the SMB in situ observations. The good performance of the PDD models in the ablation zone suggests that estimating melt from temperature remains valid under current climate conditions and that the use of more sophisticated energy balance melt scheme (like the ones used in SnowModel and NHM-SMAP) can generate larger biases than resolving them with a priori better physics. Finally, the mean bias in the ablation zone mostly explains the large discrepancy between models and GRACE-derived mass loss trend in 2003-2012. This suggests that biases over
665  current climate could strongly impact the models' ability to simulate future meltwater runoff acceleration and associated sea level rise (Fettweis et al., 2013a).

The GRACE signal variability is mainly a combination of the i) seasonal cycle (accumulation in winter and melt in summer), ii) interannual variability in this seasonal cycle and, iii) long- climate variability (linked in part to global warming) induced
670  mass loss, that we assume here to be the linear trend over the considered period (2003-2012). Over Basin 1 and 2, IMAU-ITM and SnowModel (resp. CESM2 and MPI-ESM) overestimate (resp. underestimate) the mass loss in the GRACE signal (i.e. the linear trend) over 2003-2012. Over Basin 3, all the models underestimate the mass change. Additionally, some of the models (in particular MAR) do not simulate mass variations in Basin 3, despite GRACE data suggesting a mass loss of 450 Gt over 2003-2012. In south Greenland (Basin 4), the two PDD models show the most favourable statistics but all the
675  models (except MPI-ESM) underestimate mass loss. Along the west coast (Basin 5 and Basin 6), MAR and RACMO2.3 are most closely aligned with the observations, while SnowModel systematically overestimates, and NHM-SMAP systematically underestimates the mass loss. For the other models, the bias in Basin 5 and 6 varies in sign. Finally, an EBM (dEBM), a GCM (MPI-ESM), a PDD (PDD1km) and two RCMs (MAR and RACMO2.3) compare the closest to the GRACE-derived GrIS-integrated mass loss over 2003-2012. These favourable statistics are due to error compensation as none of the models

680 matches well the GRACE-derived regional mass loss integrated over individual basins.

RCMs have the advantage that they resolve near-surface climate and dynamically downscale the precipitation to higher spatial resolution with respect to their forcing whereas the PDD and EBM models are fully driven by the near-surface climate of their low resolution forcing fields. While the two GCMs used in this study are not (or not fully) forced by ERA-Interim, they reasonably simulate the melt, that can likely be ascribed to subgrid elevation corrections applied in MPI-ESM

685 and CESM2. However, for precipitation, the native GCM resolution remains too coarse to resolve the spatial variability simulated by RCMs. The spatial variability of precipitation in RCMs is particularly high along the southeast coast of Greenland. However, the paucity of observations prevents us from confirming whether the local high precipitation rates simulated by RCMs and not captured by lower resolution models are realistic. Moreover, running RCMs at a high spatial resolution becomes computationally expensive on time scales beyond one century suggesting that the PDD/EBM and GCM

690 based approaches may be more suitable for questions that require long simulations (where a coupling to an ice sheet model may be desirable as well).

For a total surface mass loss over 2003-2012 of ~ 3000 Gt as suggested by the GRACE data set, the models range from -1066 Gt to -6034 Gt with an ensemble mean of -2611 +/- 1253 Gt suggesting a large discrepancy between models and therefore a large uncertainty in the modelled SMB trends over the current climate. It is nevertheless interesting to note

695 that, for most of the models, the sign of the bias when compared to the PROMICE data (see Table 3) is highly correlated to the sign of the trend bias with respect to the GRACE based product. For example, as the 2003-2012 changes in SMB were driven by an increase of melt, those models underestimating surface ablation also underestimate these recent changes (the signal coming from discharge change is the same for all the models). However, we need to mention that all of our modelled total mass balance estimations use the same discharge estimates from King et al. (2018) and do not take into

700 account changes in mass over tundra, over small ice caps (not included in the common ice sheet mask) and in glacial storages (e.g. meltwater lakes, water tables,...): this partly explains why the discrepancies between models and GRACE could be very high in some areas.

The different types of models show large discrepancies in regional mass loss trends with respect to GRACE estimates (2003-

705 2012) in the trend of surface mass loss between the models compared here. Comparing outputs from these same models in a future warmer climate will enable an improved quantification of uncertainties in climate projections and therefore help refine estimates of the GrIS contribution to future sea level rise.

[revised manuscript text omitted]

---

## Referee Report (RR1)

A review of "GrSMBMIP: Intercomparison of the modelled 1980-2012 surface mass balance over the Greenland Ice sheet" By X. Fettweis, et al.

This manuscript presents a surface mass balance (SMB) model inter-comparison for Greenland over the period 1980-2012. Varying types of SMB models are compared, and assessed against a number of observational datasets. The manuscript also reports a mean trend, mean, and standard deviation of SMB over the assessed period. Additionally, regional comparisons are presented, and suggest that the largest discrepancies between models are found in proximity to the ice sheet margin. Regional climate models (RCMs) compare the best against observations, especially with respect to their skill in capturing SMB components independently, but are computationally intensive. Results suggest that the less-expensive, simpler models may be appropriate for simulating SMB for longer simulations, as they have comparable biases to the RCMs. The manuscript also notes that the ensemble mean produces the best estimate of SMB compared to observations, and no systematic biases are exposed by this exercise. This inter-comparison is an important step in evaluation of the scientific community's understanding of the physical processes driving mass trends in Greenland and in the ability to determine what types of models are appropriate for answering key scientific questions, particularly with respect to future projections of spatial ice sheet change and sea level contribution.

The current revision of the manuscript is much improved over the first version. The introduction is now extremely helpful to the reader in providing details about this important inter-comparison effort and the consequences of its outcomes. The authors have appropriately responded to concerns from both reviewers, and the revisions to the tables and figures make the results and discussion more readable. For these reasons, I recommend publication in The Cryosphere with minor revisions, specifically an expansion of the discussion section.

General comments:

In general, I find that the Discussion section lacks *discussion* of scientific implications of the many interesting results presented here. Most importantly, a discussion about what the results imply about the ability of different types of models to project into the future, with consideration to the assumptions/tuning that needs to be made in order to match present-day conditions, would be appropriate. For instance, the last sentence of the conclusions implies that PDD/EBM may be better for future simulations in terms of general skill and cheaper computations, but this point is more complex than stated here. For instance, do these models have just as much skill during strong melt years (i.e. 2012), as other neutral years? Though they seem to have skill historically, are there any indications in this analysis that suggest they will do as well in an extreme future scenario? Since 2012 is the last year assessed here, a preliminary hypothesis about future skill could be made within this manuscript. If not, it would be important to state that no such conclusion can be made. Another example is the comparison against Bougamont et al., 2007, concerning PDD sensitivity (see note in comments below). Pointing out to the reader through discussion that historical results may not translate directly to skill in to future projections, as runoff becomes more important in the future, would result in the richer discussion that I would like to see presented. Currently, the discussion section, is only two paragraphs, so I suggest dedication of an additional paragraph to reflect on such questions and support the final conclusion statement of the manuscript.

Specific comments and suggestions are noted below:

Line 73, To me, the end of this statement, "but have never been evaluated until now" implies that GS is being evaluated. Earlier, the text clearly states that GS is not simulated by the models. Because GS is clearly not simulated, I do not know what "until now" refers to. Either this is a miscommunication and should be rephrased, or please be more explicit about what it meant by this sentence.

Line 90, Missing "s": model"s"

Line 92, awkward phrasing: perhaps, "the" then

Line 146, It is unclear how this exercise specifically reduces uncertainty. Please clarify why this is so or rephrase. Maybe "quantifies" could be used instead of "reduces"?

Line 196 and Line 244, Nowicki

Line 266, missing word: "the" sum

Line 290, "no" significant differences, instead of "not"

Line 382, Please include a reference to the accumulation being constant over the last decades

Lines 408-409, Does this sentence refer to the fact that in the King paper, RACMO is used to evaluate discharge estimates? The way it is written, it sounds like this current manuscript uses RACMO to compare total mass balance with GRACE. Perhaps just a simple rephrasing to clarify that you refer to the methods of the King et al., 2018 here would help alleviate this confusion.

Line 432-434, Could this be because Bougamont et al., 2007 is comparing the results of these model in an extreme warming scenario, while here, you are considering only the historical period? This point seems like something worth elaborating on with regards to SMB models and their ability to make future projections. It would strengthen the discussion to add some sentences dedicated to this subject. (See point on Discussion above).

Line 466, awkward phrasing: maybe, disallowing "the representation of" the spatial variability

Line 511, awkward: perhaps, matches well "with" (or "against") the

Line 550, It would be appropriate here, to note that these results are consistent with past SMB assessments using GRACE, e.g. "is consistent with past assessments of SMB seasonal variability using GRACE", with reference to e.g. Velicogna et al., 2014; Alexander et al., 2016; Schlegel et al., 2016

*Velicogna, I., Sutterley, T. C., and van den Broeke, M. R.: Regional acceleration in ice mass loss from Greenland and Antarctica using GRACE time-variable gravity data, Geophys. Res. Lett., 41, 8130–8137, doi:10.1002/2014GL061052, 2014.*

*Alexander, P. M., Tedesco, M., Schlegel, N.-J., Luthcke, S. B., Fettweis, X., and Larour, E.: Greenland Ice Sheet seasonal and spatial mass variability from model simulations and GRACE (2003–2012), The Cryosphere, 10, 1259–1277, https://doi.org/10.5194/tc-10-1259-2016, 2016.*

*Schlegel, N.-J., Wiese, D. N., Larour, E. Y., Watkins, M. M., Box, J. E., Fettweis, X., and van den Broeke, M. R.: Application of GRACE to the assessment of model-based estimates of monthly Greenland Ice Sheet mass balance (2003–2012), The Cryosphere, 10, 1965–1989, https://doi.org/10.5194/tc-10-1965-2016, 2016.*

Line 578, missing space: "quarters of"

Line 582-583, This is a very important statement, and the amount of tuning that each type of model does may be an important aspect of its ability to make projections in the future. I suggest touching on this idea earlier in the manuscript, with a couple of sentences of additional discussion (i.e. general Discussion comments above) to help the reader reflect on how assumptions and tuning to present day may affect future results.

Figure 4, caption – Fig. 4 refers to being the same as Fig. 23. Likely this should be Fig. 2 referred to here?

---

## Referee Report (RR2)

Review of revised manuscript, "GrSMBMIP: Intercomparison of the modelled 1980-2012 surface mass balance over the Greenland Ice Sheet" by Fettweis and coauthors.

I have examined the revised manuscript and a letter to the editor describing 8 significant changes in revision together with a tracked-change document. It should be clear that *the letter does not constitute a point-by-point response to reviewer comments*. I find that startling, and it should be noted by the editor.

From the letter and revised manuscript, it may be possible to chart progress on my seven points previously raised through some detective work, albeit without the authors' reasoning for acceding or discounting each question. In the absence of this reasoning I can only again recommend minor revision. I attempt to assess the response to the previous review below.

1. The first point asked whether the model intercomparison should be more emphasized. The manuscript appears to have been re-ordered such that the model intercomparison now appears first. This seems like a reasonable response, and it appears that the manuscript is now more focused on the intercomparison section.

2. It was suggested that the figures be re-ordered by model type as rather than alphabetically. This does not appear to have been addressed. It doesn't really make any sense to have the figures presented alphabetically, and grouping the figures by model type would better indicate the differences among these groups.

3. The third point suggested an expanded comparison with earlier intercomparison studies to provide some understanding of how models have improved. The authors do not appear to have taken up this point.

4. The abstract has been reduced in length and appears to be much improved.

5. Both reviewers commented on an overly-lengthy description of the models. The revision has added a clarifying Table 1. The introduction now also contains a section describing the four kinds of models considered. But it remains unclear as to why it is necessary to include ~300 words of description for each of the models within the main text. The specific models and most important parts of their individual characteristics should be added to the new introduction section, with the remainder of these details going into a supplementary section.

6. My sixth point asked about the difference in amount between models that computed snowfall with those that incorporated the snowfall of the reanalysis. It does appear that section 4 has been rewritten to better address questions of the model differences. Ok here.

7. A list of acronyms was suggested. The authors do not appear to have taken up this point.

---

## Author Response (AR2)

Dear Editor, Dear Robin,

Thanks for accepting our manuscript with minor revision.

In addition to the minor suggestions requested by reviewer #1, we have added this paragraph in Section 6 (discussion) discussing the importance of well simulating the current climate

*Except the two GCMs that underestimate the recent surface mass loss trend mainly because they have not the same general circulation variability than the ERA-interim climate, the runoff anomaly expressed in percentage of the ensemble summer mean remains mainly constant in time, including the extreme summer 2012. This confirms that the modelled runoff overestimation/underestimation is systematic over time. This also suggests that a runoff bias over current climate will increase in absolute value for warmer climates in the same proportion than runoff, justifying the importance of well representing the current climate before performing future projections.*

as requested by reviewer #1.

About our point-by-point response to the initial review of reviewer #2, this last one is in fact available online in the discussion of our manuscript on the TC web site. We have forgotten to remember this in our letter to you. We have added our initial point-by-point response at the end of this document and have made some comments about the 7 points raised by the reviewer in his/her 2nd review.

To conclude, we would like to thank both reviewers for their very useful comments that helped a lot to improve our manuscript.

Thanks for considering this revised version!

Best regards,

Xavier, on the behalf of all the co-authors.

Anonymous Referee #1 **(R#1)  - 27 JUNE 2020 (2ⁿᵈ review)**

A review of "GrSMBMIP: Intercomparison of the modelled 1980-2012 surface mass balance over the Greenland Ice sheet" By X. Fettweis, et al.

This manuscript presents a surface mass balance (SMB) model inter-comparison for Greenland over the period 1980-2012. Varying types of SMB models are compared, and assessed against a number of observational datasets. The manuscript also reports a mean trend, mean, and standard deviation of SMB over the assessed period. Additionally, regional comparisons are presented, and suggest that the largest discrepancies between models are found in proximity to the ice sheet margin. Regional climate models (RCMs) compare the best against observations, especially with respect to their skill in capturing SMB components independently, but are computationally intensive. Results suggest that the less-expensive, simpler models may be appropriate for simulating SMB for longer simulations, as they have comparable biases to the RCMs. The manuscript also notes that the ensemble mean produces the best estimate of SMB compared to observations, and no systematic biases are exposed by this exercise. This inter-comparison is an important step in evaluation of the scientific community's understanding of the physical processes driving mass trends in Greenland and in the ability to determine what types of models are appropriate for answering key scientific questions, particularly with respect to future projections of spatial ice sheet change and sea level contribution.

The current revision of the manuscript is much improved over the first version. The introduction is now extremely helpful to the reader in providing details about this important inter-comparison effort and the consequences of its outcomes. The authors have appropriately responded to concerns from both reviewers, and the revisions to the tables and figures make the results and discussion more readable. For these reasons, I recommend publication in The Cryosphere with minor revisions, specifically an expansion of the discussion section.

Thanks a lot for these positive comments about the revised version of our manuscript.

General comments:
In general, I find that the Discussion section lacks discussion of scientific implications of the many interesting results presented here. Most importantly, a discussion about what the results imply about the ability of different types of models to project into the future, with consideration to the assumptions/tuning that needs to be made in order to match present-day conditions, would be appropriate. For instance, the last sentence of the conclusions implies that PDD/EBM may be better for future simulations in terms of general skill and cheaper computations, but this point is more complex than stated here. For instance, do these models have just as much skill during strong melt years (i.e. 2012), as other neutral years? Though they seem to have skill historically, are there any indications in this analysis that suggest they will do as well in an extreme future scenario? Since 2012 is the last year assessed here, a preliminary hypothesis about future skill could be made within this manuscript. If not, it would be important to state that no such conclusion can be made. Another example is the comparison against Bougamont et al., 2007, concerning PDD sensitivity (see note in comments below). Pointing out to the reader through discussion that historical results may not translate directly to skill in to future projections, as runoff becomes more important in the future, would result in the richer discussion that I would like to see presented. Currently, the discussion section, is only two paragraphs, so I suggest dedication of an additional paragraph to reflect on such questions and support the final

conclusion statement of the manuscript.

We have added this paragraph:

*Except the two GCMs that underestimate the recent surface mass loss trend mainly because they have not the same general circulation variability than the ERA-interim climate, the runoff anomaly expressed in percentage of the ensemble summer mean remains mainly constant in time, including the extreme summer 2012. This confirms that the modelled runoff overestimation/underestimation is systematic over time. This also suggests that a runoff bias over current climate will increase in absolute value for warmer climates in the same proportion than runoff, justifying the importance of well representing the current climate before performing future projections.*

in Section 6 (Discussion) as well as this comment

*This suggests that the PDD/EBM and GCM based approaches may be more suitable for questions that require long simulations (where a coupling with an ice sheet model may be desirable as well), if they well simulate the current climate (in particular melt runoff).*

into the last sentence of the conclusion.

Specific comments and suggestions are noted below:
Line 73, To me, the end of this statement, "but have never been evaluated until now" implies that GS is being evaluated. Earlier, the text clearly states that GS is not simulated by the models. Because GS is clearly not simulated, I do not know what "until now" refers to. Either this is a miscommunication and should be rephrased, or please be more explicit about what it meant by this sentence.

We have changed the verb evaluated (suggesting indeed an evaluation of models) by quantified.

Line 90, Missing "s": model"s"
Line 92, awkward phrasing: perhaps, "the" then
Line 146, It is unclear how this exercise specifically reduces uncertainty. Please clarify why this is so or rephrase. Maybe "quantifies" could be used instead of "reduces"?
Line 196 and Line 244, Nowicki
Line 266, missing word: "the" sum
Line 290, "no" significant differences, instead of "not"
Line 382, Please include a reference to the accumulation being constant over the last decades

Thanks a lot for these corrections.

Lines 408-409, Does this sentence refer to the fact that in the King paper, RACMO is used to evaluate discharge estimates? The way it is written, it sounds like this current manuscript uses RACMO to compare total mass balance with GRACE. Perhaps just a simple rephrasing to clarify that you refer to the methods of the King et al., 2018 here would help alleviate this confusion.

We have removed this sentence and reformulated a bit the previous sentence to:

***Finally, in King et al. (2018),** monthly glacier discharge estimates were combined with RACMO2.3p2based SMB, and compared to the resulting total mass balance estimate from the GRACE product **as will be done in this study for each model.***

Line 432-434, Could this be because Bougamont et al., 2007 is comparing the results of these model in an extreme warming scenario, while here, you are considering only the historical period? This point seems like something worth elaborating on with regards to SMB models and their ability to make future projections. It would strengthen the discussion to add some sentences dedicated to this subject. (See point on Discussion above).

PDDs remain in the EBMs envelop (and RCMs envelop), including the extreme years like 2012. We have added this small comment (in black) to this sentence:

*Finally, while Bougamont et al. (2007) concluded that PDDs are more sensitive to climate warming than EBMs, it is not the case here as the PDD-based melt rates are fully included in the EBMs-based spread**, including over the extreme summers (e.g. 2012)**.*

Line 466, awkward phrasing: maybe, disallowing "the representation of" the spatial variability
Line 511, awkward: perhaps, matches well "with" (or "against") the

ok

Line 550, It would be appropriate here, to note that these results are consistent with past SMB assessments using GRACE, e.g. "is consistent with past assessments of SMB seasonal variability using GRACE", with reference to e.g. Velicogna et al., 2014; Alexander et al., 2016; Schlegel et al., 2016
Velicogna, I., Sutterley, T. C., and van den Broeke, M. R.: Regional acceleration in ice mass loss from Greenland and Antarctica using GRACE time-variable gravity data, Geophys. Res. Lett., 41, 8130-8137, doi:10.1002/2014GL061052, 2014.
Alexander, P. M., Tedesco, M., Schlegel, N.-J., Luthcke, S. B., Fettweis, X., and Larour, E.: Greenland Ice Sheet seasonal and spatial mass variability from model simulations and GRACE (2003-2012), The Cryosphere, 10, 1259-1277, https://doi.org/10.5194/tc-10-1259-2016, 2016.
Schlegel, N.-J., Wiese, D. N., Larour, E. Y., Watkins, M. M., Box, J. E., Fettweis, X., and van den Broeke, M. R.: Application of GRACE to the assessment of model-based estimates of monthly Greenland Ice Sheet mass balance (2003-2012), The Cryosphere, 10, 1965-1989, https://doi.org/10.5194/tc-10-1965-2016, 2016.

Thanks for this suggestion. The sentence has been reformulated and these references added.

Line 578, missing space: "quarters of"
Line 582-583, This is a very important statement, and the amount of tuning that each type of model does may be an important aspect of its ability to make projections in the future. I suggest touching on this idea earlier in the manuscript, with a couple of sentences of additional discussion (i.e. general Discussion comments above) to help the reader reflect on how assumptions and tuning to present day may affect future results.

We have added this comment at the end of the sentence:

*This suggests that biases that operate under current climate could strongly impact the models' ability to simulate future meltwater runoff acceleration and associated sea level rise (Fettweis et al., 2013a)* **as the present day melt bias increases in absolute value in the same proportion than runoff.**

Figure 4, caption - Fig. 4 refers to being the same as Fig. 23. Likely this should be Fig. 2 referred to here?

Thanks

Anonymous Referee #2 **(R#2) – 14 JULY 2020 (2nd review)**

Review of revised manuscript, "GrSMBMIP: Intercomparison of the modelled 1980-2012 surface mass balance over the Greenland Ice Sheet" by Fettweis and coauthors.
I have examined the revised manuscript and a letter to the editor describing 8 significant changes in revision together with a tracked-change document. It should be clear that the letter does not constitute a point-by-point response to reviewer comments.

Our point-by-point response to reviewer comments is in fact available in the online discussion of the manuscript on the TC web site:

https://tc.copernicus.org/preprints/tc-2019-321/

Sorry of having forgotten to remember this in our letter to the editor.

 I find that startling, and it should be noted by the editor.
From the letter and revised manuscript, it may be possible to chart progress on my seven points previously raised through some detective work, albeit without the authors' reasoning for acceding or discounting each question. In the absence of this reasoning I can only again recommend minor revision. I attempt to assess the response to the previous review below.
1. The first point asked whether the model intercomparison should be more emphasized. The manuscript appears to have been re-ordered such that the model intercomparison now appears first. This seems like a reasonable response, and it appears that the manuscript is now more focused on the intercomparison section.

See our comments in our responses to the initial review.

2. It was suggested that the figures be re-ordered by model type as rather than alphabetically. This does not appear to have been addressed. It doesn't really make any sense to have the figures presented alphabetically, and grouping the figures by model type would better indicate the differences among these groups.

As justified in our response to the initial review, we initially thought to group together models by type but we prefer to let the legend sorted alphabetically, because the spread inside a model type is of the same order as the spread over all models and because model type has not systematic bias. Therefore, there is no raison to put together models by type. Finally, the fact that we can not make a conclusion about a specific type of models is several times highlighted in the manuscript.

3. The third point suggested an expanded comparison with earlier intercomparison studies to provide some understanding of how models have improved. The authors do not appear to have taken up this point.

As detailed in our responses to the initial review, a large part of model spread found in Vernon et al. (2013) was only due to the fact the ice sheet mask was larger in some models. As we use the same ice sheet mask for each model, our results are therefore not comparable to the Vernon et al. (2013) ones. Finally, the intercomparison of Bougamont et al. (2007) is discussed in the manuscript (Lines: 433-435)

4. The abstract has been reduced in length and appears to be much improved.

Thanks, with the help of the initial reviewers' suggestions!

5. Both reviewers commented on an overly-lengthy description of the models. The revision has added a clarifying Table 1. The introduction now also contains a section describing the four kinds of models considered. But it remains unclear as to why it is necessary to include ~300 words of description for each of the models within the main text. The specific models and most important parts of their individual characteristics should be added to the new introduction section, with the remainder of these details going into a supplementary section.

As Reviewer #1 requested more details about the models, we think that the current description of a model is a good compromise. In addition, as some models have not yet an accepted publication about the model version used in this intercomparison, a brief description of these models is needed in the main manuscript and by extension, of all the models.

6. My sixth point asked about the difference in amount between models that computed snowfall with those that incorporated the snowfall of the reanalysis. It does appear that section 4 has been rewritten to better address questions of the model differences. Ok here.

Thanks

7. A list of acronyms was suggested. The authors do not appear to have taken up this point.

All the acronyms are listed in Table 1 and/or are summarised in the legend of Table 1. We do not think that an additional table is needed here.

Anonymous Referee #1 **(R#1)** - **28 FEBRUARY 2020 (1ˢᵗ review)**

The following is a review of "GrSMBMIP: Intercomparison of the modelled 1980-2012 surface mass balance over the Greenland Ice sheet" By X. Fettweis, et al. The manuscript presented describes the Greenland Ice Sheet surface mass balance (SMB) model inter-comparison results for the historical period 1980-2012. The authors assess the ability of different types models (including regional climate models, radiation balance models, positive degree day models, and general circulation models), 13 in all, to estimate the surface mass balance over the Greenland continent. Skill criteria for the models are derived from observational datasets, including MODIS bare ice extent, ice cores/snow pits/ in-situ observations, and the calculation of regional SMB as the C1 difference between GRACE estimates of mass and previously-published ice discharge into the ocean. A large amount of effort is taken to design the experiment and compile model submissions that have the same source of forcing, the same spatial resolution, and cover the same overlapping time periods. Through this comparison, the authors derive an ensemble mean and standard deviation of Greenland SMB over the 1980- 2012 period, as well as trends. The authors find the largest model discrepancies are along the ice sheet margins, and it is the increase in meltwater runoff along the margins that drive the prevailing negative trend in Greenland mass balance over the study period. Results suggest that regional climate models have strong skill in matching observed patterns of SMB, though computationally expensive compared to the positive degree day or radiation balance models. Overall, the authors find that it is the ensemble mean that best matches observations, meaning that errors from the various models balance each other out and do not convey any obvious systematic biases.

The work presented here is critical for cryosphere scientists, especially to the scientists interested in quantifying and simulating the evolution of ice sheets (including atmosphere/surface/ice sheet modelers). This is clearly a massive effort, and as observation of SMB in many areas are quite sparse, the authors have done a very nice job of compiling meaningful comparison criteria as a first attempt at this type of exercise. Such an effort is quite necessary to build a SMBMIP community and launch similar efforts in the future. The work presented here is especially a nice basis on which to build future comparison efforts that may focus on sea level projections. This is especially true considering the conclusion that the current compilation of models does not show systematic bias. For these reasons, publication of this work is timely and critical.

Thanks for these comments.

That being said, the manuscript in its current form needs a lot of work, especially the text which requires major revision. The tables and figures, in general, are adequate for conveying the discussion and conclusions of the manuscript. However, the model descriptions take up most of the text, and the rest is very concise. I think expanding upon the scientific results would make this manuscript much less of a technical paper C2 and much more appropriate for publication in The Cryosphere. Such improvements would also help broaden the audience for this paper. As is, the manuscript is difficult to digest by other cryosphere scientists, and the authors do not make it immediate clear to the reader why these impactful results may be of interest to their research.

Below, I outline my general comments to the authors:

**R#1 1. Introduction**
**R#1** 1.1 In general, the introduction should be expanded to discuss more clearly the topics of observed variability in SMB over the historical time period assessed and why it is important and/or difficult to capture them with models. In addition, an introduction to the types of models that are

assessed should be given, since those reading this manuscript might not be familiar with how and why these types of models differ. This could be a good way to let the reader know about the general advantages and disadvantages of each model type also. Another helpful topic to cover would be a short discussion on why this effort is so important and what the authors are aiming to learn about model bias (i.e. why a historical assessment is helpful to complete before assessing projections from this group of models). This pertains to statements that are made in the conclusion section of the paper, especially those about implications on model coupling and about quantification of uncertainties in sea level rise projections. Introducing these concepts before mentioning them in the concluding remarks would help highlight their importance and future inter-comparison goals.

We fully agree with these suggestions and we plan to improve our introduction following them.

**R#1** 1.2 The first paragraph of the introduction mentions glacial water storage, and notes that this is the first time it has been evaluated. However, the GS term is not included in Equation 1, and GS is not discussed explicitly anywhere else in the manuscript. Please be more specific here about how GS is evaluated, and how it is being included in this analysis.

Sorry that our sentence aiming to tell that GS is not considered here was ambiguous. We will therefore rewrite our sentence to
*Moreover, as it is simulated by no model considered here, the water glacial storage (GS; lakes, melt pond, channels,…) is neglected in this intercommunion although the mass changes coming from GS, when SMB is integrated over the whole ice sheet, could be relevant (but has never been evaluated until now).*

**R#1** 1.3 Line 70-71, This line might be better coming after Eq. 1. C3

OK. GS is also missing in Eq. 1

**R#1** 1.4 Line 78, Please specify the type of variability you refer to here

OK. It is the surface water runoff increase.

**R#1 2. Model Section**
**R#1** 2.1 Maybe it would be helpful if there was an overarching Methods section, since the Model section would really benefit from a short introduction describing what your methods in general are, and what you plan to do as an inter-comparison exercise. If the Model, Observations, and Evaluation all came under a larger section, it might be a good way to add some explanation prior to the reader going through all the details right away without understanding what type of inter-comparison is being presented.

Such a section will be added at the end of the introduction.

**R#1** 2.2 As it is, it is very difficult and quite boring for the reader to be introduced to a list of models and their descriptions up front with no introduction to them. Maybe a table of model names, types, native resolution, downscaling type, etc., could help serve as a reference/summary to this section. Such a table/figure might help the reader to have something to refer to while looking through the tables and the figures. Easier access to model type (by a table or color coding in the figures?) would help make the results easier to read.

This is an excellent suggestion. A table summarising all of this information will be added at the

beginning of Section 2.

**R#1** 2.3 If at all possible, it also might be helpful to push this list deeper into the section
**R#1** 2.4 maybe with the observations or data described first? (Though this might be fixed by section summary I mention earlier). - It is also important to note, that many model descriptions refer to RACMO within their write-ups, but no reference for RACMO, what it is, or what it stands for has been included prior to these sections.

As reviewer #2 recommend to put the model inter-comparison (Section 5) before the evaluation of models (Section 4), we prefer to leave the order of Section 2 and 3 as it as the comparison with data will be discussed after the model.
The new table at the beginning of Section 2 will mention RACMO allowing the reader to better know what is RACMO before describing it afterwards more in depth.

**R#1 3. GRACE estimation Section**
**R#1** 3.1 It could be helpful to include the equation of glacier mass balance here, so that it is clear to the reader how SMB is calculated from GRACE and ice discharge.

OK, the equivalent of Eq 1 for SMB will be added.

**R#1** 3.2 These last two sentences can probably be simplified to just say that you are using the methods of King et al. (2018), but instead of RACMO, you use each of your different C4 SMB products. These sentences are awkward that way that they read currently. Comparison with GRACE measurements Section

Excellent suggestion. We agree that our last two sentences are not very clear.

**R#1** 3.3 Results here are very interesting and there is plenty to point out to the reader. In general, I don't think there is any advantage to being extremely concise. It would be nice for you to lead the reader from figure to conclusion for some of the statements made in this section.

We plan to add figures similar to the one shown below, discussing the mean SMB rate/trend simulated by the models vs the mean bias with GRACE. This figure notably shows that the models with the largest SMB rate (due to higher snowfall or lower runoff) systematically underestimates the recent GRACE derived surface mass loss.

[Figure]

**R#1** 3.4 The discussion about the seasonal cycle here is a nicely suggested by your presented results. Could you please add some more explanation in order to lead the reader a bit more on why the RMSE from your Supp. Fig. 2 would imply how the seasonal cycle is modeled?

OK. We will better explain in the revised version that removing the trend allows us to evaluate the seasonality of the signal (which is a combination of both the seasonal cycle and the Global Warming induced mass loss).

**R#1** 3.5 For discussing the GCM's, could you please be explicit about the difference between the forcing of variability on these models by ERA-Interim, and how it pertains to the RMSE?

OK. Idem. We agree that the problem of GCMs not using ERA-Interim as forcing is not sufficiently explained in-depth.

**R#1 4. Conclusion Section**
**R#1** 4.1 Here, some of new concepts that were not brought up earlier in the manuscript are mentioned. This includes the mention of coupling with an ice sheet model (i.e. a topographic feedback scheme was not used, maybe add a reference to a paper that shows the feedback may be important) and quantification of uncertainty in climate projections. I think the manuscript would be improved if some space was taken in the discussion section to mention more of how results presented here do have implications for these other applications. Implications of interest could

range from estimates of historic sea level contribution to forcing of ice sheet models for historic and future simulations, and the ability to now give those applications error bars. I would even say that bringing these applications up in the Introduction as justification for conducting this MIP could help improve the manuscript's impact.

OK.

**R#1 5. Below, I offer some more specific comments/suggestions:**

Thanks for all of these suggestions that will be taken into account in the revised version of our manuscript.

Page 2, line 62: Please rephrase, "of the same order as RCMs compared with observations and therefore remain useful tools. . ." or something similar C5
Page 3, line 98: maybe, "although each model prescribes the reanalysis forcing in different manner". Please refrain from referring to the forcing as data.
Page 3, line 99: "(EMBs)"
Page 11, lines 1-2: A reference to Fig. 1 would be helpful here
Page 11, line 336-337: "This allows. . ." Please rephrase this sentence. It is very awkward and difficult to understand.
Page 12, line 359: maybe, ". . .compared to the resulting mass balance estimates from the GRACE product".
Page 13, line 411: "with the GRACE-derived. . ."
Page 15, line 452: Instead of mainly, maybe "largely"?
Page 15, line 457: Not sure what you mean by "oscillates" in this context. Maybe "deviates from the mean"?

Anonymous Referee #2 **(R#2) – 19 MARCH 2020 (1st review)**

The manuscript presents an experiment in which the surface mass balance (SMB) output of five regional climate models, four surface energy balance models, and two positive degree day (PDD) schemes for the Greenland Ice Sheet (GrIS) are each forced with ECMWF-Interim atmospheric reanalyses over the period 1980-2012. They are compared with each other, with available in situ observations, with MODIS-derived bare ice extent, and with a derived gravimetric data set in which an observed terminal glacier discharge has been incorporated. The output from two global general circulation models is also considered. The main results presented are that the models simulate a statistically significant decrease in SMB over the period, that the largest differences between models occur on the ice sheet margins, and that regional climate C1 models generally perform well in comparison to the validation data.

The manuscript is around 7500 words with 6 figures and 4 tables, which is a reasonable length for the topic. It represents a considerable community effort in organizing and executing the experiment. The author list comprises most major modeling efforts for contemporary GrIS SMB with the outstanding exception of atmospheric reanalyses (e.g., the Arctic System Reanalysis; MERRA-2). The initial reaction is that this is a significant update on earlier efforts of Vernon et al. (2013) and perhaps Rae et al. (2012) in model assessment. While those studies were mostly focused on regional climate models, this manuscript aspires beyond that with the inclusion of a large number of surface energy balance and PDD models. I have a few points for the authors to consider below, and so would suggest some revision of the manuscript.

Thanks for these comments.

**R#2** 1. The experiment necessarily relies on the common use of one forcing product, ECMInterim. By itself, this study is then not a complete characterization of SMB and its uncertainty from model sources, as the uncertainty of the forcing would also need to be considered. The use of different forcing products is beyond the scope of this study, but it would seem that the forcing selection plays a significant role in determining trends. Consider that if one wished to comprehensively evaluate a forcing product for the GrIS, a possible approach would be this experiment: a comparison of many forced models with observations may be seen as an elaborate validation of the forcing product. Is that not so? The purpose of this study is an appraisal of the different models, and for this purpose the key results are in how models compare with each other, and the systematic differences between them. These would seem to be the results that should be emphasized. Comparisons with observations are of interest (e.g., Fig. 1 is very interesting) but would not seem to be the principal outcome to be emphasized. The manuscript presents a considerable amount of information on the intercomparison in the form of figures and tables. Beginning in section 4, the text focuses primarily on the direct comparison with observations. The intercomparison is largely covered in the second paragraph of section 5. It is suggested that the results be re-ordered with the C2 intercomparison presented first. Some additional aspects of the intercomparison may be highlighted, as suggested below.

As suggested to reviewer #1, the models inter-comparison (Section 5) will be put before the evaluation of models (Section 4) in the revised version. We agree with the comments of the forcing (ERA-Interim) vs observations. However, according to Fettweis et al. (2017) who forced MAR with 6 reanalyses, the impact of the forcing on the modelled results over the recent decades (in particular over 1980-2012 considered here) remains negligible with respect to the model discrepancies we found here over this same period. The modelled results' dependence on the forcing as well as the associated impacts on the comparison with observations will be taken into account in the revised

version of our manuscript.

**R#2** 2. For tables and Figs. 2, 4, 5, and 6, the plots should be sorted by model type rather than alphabetically, and perhaps labelled accordingly. It may also be useful to plot the spread for each model type.

Initially, we thought to do this but we prefer to show the legend alphabetically sorted and not put models of the same kind together, because the spread inside a model type is of the same order as the spread over all models.

For example, the spread around the mean SMB, snowfall and runoff listed in Table 4 is generally of the same order for the total of 13 models than for a class of model, except for the 2 PDD models, which are very similar in their design and underlying assumptions (except the resolution).

|  | Std dev around the mean SMB | Std dev around the mean Snowfall | Std dev around the mean runoff |
|---|---|---|---|
| EBM (4) | 131 | 43 | 144 |
| RCM (5) | 32 | 73 | 117 |
| PDD (2) | 33 | 11 | 34 |
| GCM (2) | 96 | 78 | 43 |
| Total (13) | 91 | 81 | 109 |

**R#2** 3. It is useful to continually compare this experiment with the previous efforts cited in the introduction. Vernon et al. found SMB estimates were within 34% of the multimodel mean of 4 models. Table 4 suggests this value is now something like 22% for the 5 RCMs but close to 100% when all of the models are considered. Does this suggest an increasing proficiency within the RCMs. Also, it is noticeable that the manuscript does not indicate surface temperature sensitivity. It is difficult to include and assess the PDD and EBM models without that consideration. For example, this was a focus of Bougamont et al., who found that PDD models were more sensitive than EBM models. Given the same forcing and the trends shown in Table 2, it does not appear that a similar conclusion holds here, is that correct?

We think that the largest difference between the 34% found by Vernon et al. and the 22% shown here is the use of a common grid and ice sheet mask. As the models in Vernon et al. did not use the same ice sheet mask, a great part of the spread around the mean was only due to the fact the ice sheet mask was larger in some models. Therefore, the difference between the models presented here can not be compared like-for-like with the differences shown in Vernon et al. (2013).

Indeed, the trend in runoff shown in Table 4, driven by the temperature increase from the end of the 1990's, is generally lower for PDDs than for EBMs except BESSI and RCMs. It is nevertheless important to note that the increase of solar radiation (not taken into account in the PDD) has also played an important role in this meltwater increase (Hofer et al., 2017).
As explained in Fettweis et al. (2013), the recent and future changes are more sensitive to the ability of the models to simulate the current mean runoff, independently of the formulation used as the melt does not increase linearly with temperatures. About GCM, this trend is also lower, mainly because they do not simulate the general circulation changes in summer as recently observed (Hanna et al., 2018) and driving in part the recent surface meltwater runoff.

*Hanna, E., Fettweis, X., and Hall, R. J.: Brief communication: Recent changes in summer*

*Greenland blocking captured by none of the CMIP5 models, The Cryosphere, 12, 3287–3292, https://doi.org/10.5194/tc-12-3287-2018, 2018.*

This issue will be discussed more in-depth in the revised version of our manuscript.

**R#2** 4. At 445 words, the abstract is too long by half. A large part of the abstract is devoted to motivation, which should instead be mostly left to the introduction. As suggested in the previous point, may consider adding more text regarding the resulting differences between the models.

Ok, thanks for these suggestions. The abstract will be shortened.

**R#2** 5. A concern is the very lengthy description of the models contained in section 2. The descriptions include sub-model components, the forcing time scale, vertical resolution, ancillary forcing data, etc. It is of course useful for close examination of individual model results, but this is generally available elsewhere from the cited literature, and it is not clear that all of it is directly pertinent to the aggregate results presented for understanding cryospheric modeling. It is suggested that this material may be incorporated into supplementary text and/or condensed with a table that includes model type, references, and major points. Otherwise it could be argued that this type of material is C3 more appropriate for a publication such as Geoscientific Model Development.

As requested by Reviewer #1, a table listing model name, type, resolution, forcing, … will be added at the beginning of Section2. As Reviewer #1 tends to request more details about the models, we think that the current description is a good compromise.

**R#2** 6. Lines 455 and following. As indicated, it is apparent from Fig. 6 that the snowfall from the EBM models, which is directly imported from the forcing, is low (mostly blue) compared to the mean over the interior regions of the ice sheet. Would it be correct in saying that models that compute snowfall generally show higher amounts than the forcing? This appears to be true for most of the HIRHAM, NHM-SMAP, and RACMO RCMs and to some extent for the BOX13. Is that an expected systematic response?

Yes and no.
Ettema at al. (2009) found that higher the resolution, higher the simulated precipitation with RACMO is but it is no more the case with the more recent versions of RACMO (Noël et al., 2019). Moreover, Franco et al. (2012) found that lower the resolution is, higher the simulated precipitation with MAR in the interior of the Greenland ice sheet is, mainly because the topographic barrier effect is less efficient in the MAR model. We think that these differences are more driven by the physics of the models and by the different downscaling methodologies + corrections (eg. for PDD) applied to the ERA-Interim based forcing data.

*Ettema J and 6 others (2009) Higher surface mass balance of the Greenland ice sheet revealed by high-resolution climate modelling. Geophys. Res. Lett., 36(12), L12501 (doi: 10.1029/2009GL038110)*

*Franco, B., Fettweis, X., Lang, C., and Erpicum, M.: Impact of spatial resolution on the modelling of the Greenland ice sheet surface mass balance between 1990–2010, using the regional climate model MAR, The Cryosphere, 6, 695–711, https://doi.org/10.5194/tc-6-695-2012, 2012.*

*Noël, B., van Kampenhout, L., van de Berg, W. J., Lenaerts, J. T. M., Wouters, B., and van den Broeke, M. R.: Brief communication: CESM2 climate forcing (1950–2014) yields realistic Greenland ice sheet surface mass balance, The Cryosphere Discuss., https://doi.org/10.5194/tc-2019-209, in review, 2019.*

**R#2** 7. A list of acronyms in the appendix would be useful.

OK, we will add this.

[revised manuscript text omitted]

---

## Author Response (AR3)

Dear Editor, Dear Robin,

Please find hereafter a revised version better highlighting the discussing the models' performance under warmer climates.

The main changes concern Lines 561-567 and Lines 589-596.

Thanks for considering this revised version!

Best regards,

Xavier, on the behalf of all the co-authors.

[revised manuscript text omitted]